# Venues and segregation: A revised Schelling model

Daniel Silver[1][☯], Ultan Byrne[2][☯], Patrick Adler[3]*

**1** Department of Sociology, University of Toronto Scarborough, Toronto, Ontario, Canada, **2** Graduate School of Architecture, Planning, and Preservation, Columbia University, New York, New York, United States of America, **3** Luskin School of Public Affairs, UCLA, Los Angeles, California, United States of America

☯ These authors contributed equally to this work.
* patrickadler@rotman.utoronto.ca

**Data Availability Statement:** All videos from the simulations are available on YouTube. The link contains the minimal data set (https://www.youtube.com/channel/UC9wuOlAb5aS90le1L_R-Y0w)).

## Abstract

This paper examines an important but underappreciated mechanism affecting urban segregation and integration: urban venues. The venue- an area where urbanites interact- is an essential aspect of city life that tends to influence residential location. We study the venue/segregation relationship by overlaying venues onto Schelling's classic (1971) [1] agent-based segregation model. We show that a simulation world with venues makes segregation less likely among relatively tolerant agents and more likely among the intolerant. We also show that multiple venues can create spatial structures beyond their catchment areas and that the initial location of venues shapes later residential patterns. Finally, we demonstrate that the social rules governing venue participation alter their impacts on segregation. In the course of our study, we compile techniques for advancing Schelling-style studies of urban environments and catalogue a set of mechanisms that operate in this environment.

## Introduction

Uncovering the structural bases of residential segregation has been among the central tasks of urban studies. Researchers have identified a number of contributing factors. These include the distribution of employment opportunities [2–5]; legal exclusion and discriminatory access to housing [4, 6]; the prevalence of minority populations and the corresponding clustering and majority reactions this may produce [7–11]; street layouts and other physical boundaries [12–15]; historical norms governing personal interaction [16]; city age [17, 18]; and more. Recently, the conceptual scope of segregation research has widened to include activity space [19, 20] and workplaces [21–24].

In this paper, we identify and examine the implications of a relatively understudied mechanism: physical venues, such as churches, cafes, restaurants, workplaces, and shopping malls. We draw inspiration from a pervasive fact of urban life, namely, that we spend much of our time in public and this interaction takes place in buildings. While this is an obvious truism, we submit that it holds far-reaching but relatively underappreciated implications for understanding the generative processes that result in patterns of residential segregation and integration. Accordingly, we propose several key variables through which physical venues shape segregation, such as their number and distribution, the degree of obligation they impose upon individuals, their catchment areas, and their norms of social exclusion and inclusion.

**Funding:** The author(s) received no specific funding for this work.

**Competing interests:** The authors have declared that no competing interests exist.

To evaluate the possibility that physical venues produce distinctive and recognizable forms of urban segregation and integration, we developed an agent-based model (ABM). ABMs have become increasingly important in the social sciences, as they allow researchers to isolate potential causal mechanisms of social phenomena and perform conceptual experiments to investigate how hypothesized mechanisms generate outcomes under varying conditions [25–28]. Their value comes not in making detailed predictions but in pinpointing how classes of phenomena in principle can generate different types of outcomes, while contributing toward an expanding and increasingly refined collection of mechanism-based explanations [29].

Ever since Schelling's classic work, "Dynamic Models of Segregation" [30], ABMs have informed research into residential segregation (for example Benenson, Omer, and Hatna [31]). Schelling demonstrated that one cannot simply read individual intentions from collective patterns of behavior. A highly segregated order could have emerged from the repeated interactions of relatively tolerant individuals. In this way, Schelling's model works as a "proof of concept" that offers an invaluable lesson into a deep irony of social life, in the sense articulated in Brown [32: 184]: a turnaround or reversal whereby agents, by acting on their intentions, produce outcomes contrary to those very intentions.

In Schelling's original model, individuals make location decisions based on the composition of their neighborhood, seeking locations where enough members of their own group are present. This paper adds physical venues to Schelling's model of individual decision-making. Methodologically, this means constructing a model in which agents' evaluation of their location includes *both* nearby residents *and* other agents they encounter in the venues they visit. This addition in turn allows us to formally model how venues modify and channel the clustering processes Schelling identified. Our guiding hypothesis is that venues can account for a crucial aspect of residential segregation that is missed in an individual-level model, namely, that segregation and integration are place-based: they recur in distinct locations, giving rise to common urban patterns such as East and West side, core and periphery, or ethnic neighborhood. These topologies should be understood as structures unto themselves that can constrain or potentiate segregation outcomes.

Guided by hypotheses drawn from reviewing and synthesizing a scattered but illuminating literature on venues, our analysis proceeds through a series of three simulation case studies of increasing sophistication. Throughout, we return to our central proposition, namely that the segregation and integration patterns observed in the venue-less Schelling world can be completely overturned once venues are introduced. In our first studies (1 and 2), venues are treated as historical artifacts that belong to one group or another. We investigate the impact of the presence, number, and distribution of venues on segregation and integration patterns. We examine abstract forms of familiar spatial configurations of venues in the urban studies literature [33–36], such as a radial and a core and periphery model. We show that at relatively low levels of intolerance, the introduction of venues can lead to more integration than would be predicted in the Schelling model, but that at higher levels of intolerance the segregated outcome becomes more likely.

In our third study (3), we consider the hypothesis that rule changes at the venue level can affect the level of mixing at the local/neighborhood level. This amounts to a study of how planning decisions can come to influence the individual decisions of Schelling-type residents. Our study focuses on how variations in a venue's *exclusivity*–the extent to which venues of a given group are open to admitting members of other groups—impact the mixing of agents. We show that more exclusive venue formats lead to more global segregation as agents migrate to be close to venues that they identify with, but that the level of segregation around venues is not linearly related to venue exclusivity. These results demonstrate that variations in rules operating at the venue scale can indeed affect higher-order residential patterns.

In the course of testing our core propositions about the role of venues in modifying Schelling-style segregation outcomes, we uncovered a number of generative mechanisms that produce the aggregate patterns we observe. Our conclusion collates and codifies these mechanisms uncovered in the course of our research. Building up toward and compiling this collection of mechanism-based explanations is one of the key contributions of this paper. The conclusion also reflects upon limitations of our model and additional work that could extend the insights from the paper further.

## Literature review

In this paper, we develop a computer simulation model that critically engages Schelling's "Dynamic Models of Segregation" (1971), seeking to test the impact of venues in generating distinctively place-based patterns of residential segregation and integration. To provide background and motivation for our modeling approach, we 1) summarize Schelling's model, 2) discuss some of its features that limit its capacity to model distinctively urban processes, namely that it includes neither public interaction nor historical memory, 3) propose incorporating venues as an austere way to include these in the model, and 4) elaborate the concept of a venue.

### Schelling simulation models of segregation

Schelling [37] is a *locus classicus* for a fundamental social scientific insight: individuals' micromotives can generate collective macrobehaviours that need not reflect (in a straightforward way) their intentions. In this case, individuals motivated by a relatively high degree of tolerance produce highly segregated "neighborhoods," even if this is not what they intend. To demonstrate this, Schelling describes an experiment involving a random distribution of simulated individuals from two distinct groups (which here will be represented as Red and Blue) in a two-dimensional grid of cells. Fig 1 summarizes the basic logic of the classic Schelling model.

Schelling assigns agents simple threshold-based behavior: if the ratio of like-neighbors (where likeness refers to shared group membership) to total neighbors is below some value, the agent will attempt to relocate to an empty cell where this measure is satisfied. For Schelling,

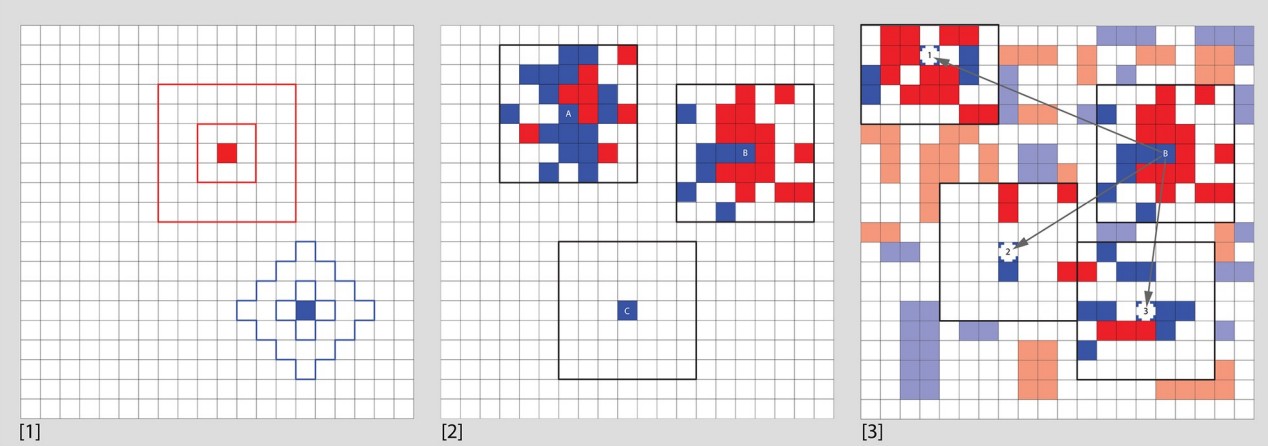

**Fig 1.** [1] Moore and Von Neumann neighborhoods at D = 1 and D = 3. In this paper, we use Moore neighborhoods consistently. [2] Three simulation agents: 'A' has 16 like-neighbors out of 24, 'B' has 6 like-neighbors out of 21, and 'C' has no neighbors. With an intolerance threshold of 0.35, 'B' is forced to move, while 'A' and 'C' remain at their current locations. [3] In a more populated world, 'B' is deciding where to move. Of the locations 'B' considers, '1' and '2' will not satisfy its intolerance threshold, but '3' will.

this threshold-value represents the relative tolerance or intolerance of the agent towards members of the other group. The "neighborhood" within which tolerance is measured is defined as the surrounding cells up to some maximum distance (whether as a Moore or Von Neumann measure) from the agent's own location. If an agent is compelled to move based on their current neighborhood, they will consider moving to available grid cells (closest first for Schelling but random order in our studies, in keeping with the subsequent literature: [38, 39]) until they find an appropriate new location. Note that this random movement means that agents do not necessarily seek out the ideal or maximally satisfying location; instead their behavior is satisficing. If none of the available cells is satisfactory, the agent will remain at their current location, looking for an appropriate destination each turn or until the makeup of their neighborhood changes. The movement of each agent can prompt moves by other agents in the neighborhood of its previous location, its new location, or both.

Schelling's key insight was that based only on this local and simple decision-making procedure, larger scale patterns of behavior appear. He establishes tolerance values at which these macro patterns are not intuitively linked to the micromotives of the agents. For example, after a series of moves, a relatively tolerant initial population (comfortable with as little as one-third of their neighbors being of the same group) will resolve itself into segregated patterns at the overall scale of the simulation world (see S1 Video). Equally crucially, once these segregated patterns emerge, they are very difficult to unsettle, at least within the parameters of the model.

## The limits of the Schelling model: Interaction and history

Subsequent research has tended either to systematically investigate simulation outcomes by varying the model's input values, or to expand on Schelling's work by introducing altogether new parameters. Some key innovations upon which we build include Fossett and Dietrich's (2009) thorough testing of the robustness of Schelling-style outcomes to a variety of inputs; Laurie and Jaggi's study of the agents' concept of neighborhood distance or "vision" [40]; and Wasserman and Yohe's introduction of "public goods" in their variation on the classic simulation [41]. The latter in particular points toward our examination of a third kind of cell, in addition to empty cells and those occupied by individuals.

A key insight that emerges from examining this tradition is that the setting for Schelling's models lacks public interaction and history. Regarding interaction, each residence is private and ruled by fiat. Owners make independent decisions guided by their private valuations and the information they have about their neighbors. Their utility consists in the satisfaction of knowing that they have a minimum number of similar neighbors. There is no interaction nor is there public space in which to interact. This austere setting is rich enough to highlight how spatial structure emerges from micro-behaviors but does not permit analysis of segregation in a recognizable urban environment.

The core Schelling model also has no history or memory. In the Schelling model, the blue agent's arrival to a location makes it fully a blue place from the standpoint of their neighbors. There is no sense of architectural or cultural legacy by which past residents continue to maintain a local presence. Similarly, there is no deliberate planning to retain a particular configuration. Agents leave when they become uncomfortable with their neighbors. They do not seek to control the future, either by exerting effort to exclude outsiders or actively seeking to include them.

## The value of incorporating venues

Incorporating venues into the Schelling model offers an austere and novel way to examine the role of interaction and history within a recognizably urban framework. In simplest terms, this means adding venues to the Schelling agent's evaluation of their location. In this approach, the

traditional Schelling model is a special case in which agents only consider their neighbors but ignore those they encounter at venues such as office, work, school, church, and the like.

There are several important theoretical reasons for making this addition. First, venues are where urban actors interact. "Venues" include firms, schools, stores, bars, clubs, parks, airports, and churches. They are where urbanites access the benefits of urban agglomeration: sharing resources, engaging in deep divisions of labor, learning new ideas [42, 43]. They are also where heterogeneous groups confront each other, potentially establishing distinctly urban and cosmopolitan social relations [44, 45]. For as long as there have been cities, there have been venues. The ancient city had its markets, agoras, temples, and palaces. A venue-less city is hard to fathom, and therefore it is important to incorporate venues into models of urban segregation.

Second, incorporating venues permits us to model backward and forward-looking processes. Urban venues can be temporally behind or ahead of their populations. From behind, there is the example in many American communities of the old Italian or Polish church in a majority-Latino community, a vestige of the previous immigrant community that continues to shape the present neighborhood [46, 47]. Looking forward, there is the first restaurant concept of what will be a popular fad and attract new groups to the area [48, 49]. Introducing venues allows us to model these processes.

Third, building a venue-based model is important because venue concerns are, empirically, a major consideration in how households locate. The good school [50], the nearby workplace [51], the cool bar [52], and the grocery store with the right milk [53] are all common signposts for location decisions. Real estate advertisements emphasize these venue cues even as they give subtle cues about what groups neighbors belong to. Just as important is the fact that the venue question will interact with Schelling's neighbor question. If we make the reasonable assumption that the venue and the neighbor's house can't occupy the same place, then marginal proximity to one tends to imply distance from the other; the agent's location in space will tend to be a kind of compromise. Even the most pious Catholic will have to decide if they want to be closer to the church or the parishioner. The addition of the venue makes the Schelling agent much more like the vendor on the beach in the classic Hotelling [54]. Study, where agents must maximize their position relative to their competitors and customers. This two-dimensional decision framework calls into question the stability of traditional Schelling-type results. For example, does access to an in-group venue make a higher number of out-group neighbors tolerable? Is proximity to an outgroup venue repulsive to intolerant residents? Incorporating venues permits us to pose such questions to gain a better understanding of how Schelling-like processes work in cities.

*What makes a venue a venue*? While the general relationship between space and society has animated a long tradition of urban research [55], the more narrow question of the impact of venues on segregation patterns is relatively underexamined within the urban residential segregation literature. Nevertheless, we can draw upon existing studies from disparate discussions of venues to help identify and elaborate theoretically important features to include as parameters in our simulation framework. Our review identifies four: venues' *physical* features (such as their number and distribution); their *catchment area*; their *mandatoriness* for individuals; and their degree of *openness*. Identifying and elaborating these parameters is important for building a conceptual framework for studying venues. In order to demonstrate its core proposition about the power of venues to alter Schelling-style segregation dynamics, however, the present paper's experiments only explore some of these parameters, leaving deeper examination of others to future research.

1. <u>Physical features.</u> A first important feature of venues that emerges in prior studies concerns the simple presence of physical venues, as well as their number and distribution. For

example, Gieryn's influential work [56] on "What Buildings Do" highlights how buildings stabilize social life by providing focal points at which social interaction regularly occurs. In a similar way, Menchik [57] brings together several strands of ethnographic research on the importance of venues [e.g. 58–62]. In this tradition, a venue "is a place people visit repeatedly to recognize the same problems, do the same tasks, and achieve the same kinds of solutions" ([57]: 853). As such, they "tug" people into (or away from) certain interactions, thereby anchoring relationships in particular physical locations, around which regular patterns of social organization revolve. In addition to concurring that venues organize social life in virtue of being foci for social interaction, Graziul [63] notes that venues are finite. This produces constraints on how interaction can occur, based on the number and distribution of venues. A city with one church, one gas station, and one grocery store, all located at the center of town, provides very different opportunities for interaction than one with dozens or hundreds of venues distributed throughout multiple neighbourhoods. To capture this theoretically important aspect of venues, our simulation framework allows researchers to examine variations in the number and location of venues.

2. Catchment area. Research has also noted how venues affect segregation through their catchment areas, or the distance from which visitors travel to attend them. A venue with a larger catchment area can affect areas of the city far away from its local coordinates. Some ethnographic work offers illustrative examples of this process. For instance, McRoberts' [64] study of clusters of storefront churches in Boston revealed the extent to which their membership was drawn from throughout the city. Similarly, Hackworth and Rekers [65] show that in the case of Toronto's Greektown, many Greek-Canadians have continued to visit Greektown churches, restaurants, and shops well after they have moved to other, often suburban, areas. In both cases, the venues' wide catchment areas helped to produce more mixed neighborhoods in other areas of the city: individuals sustain connections with members of their own groups by visiting relatively homogenous venues outside their own neighborhoods, which in turn provided a solidaristic anchor that increases openness to living in a more heterogeneous area elsewhere. Given the theoretical importance of catchment areas, our simulation framework includes a parameter to capture the distance agents are willing to travel to visit a venue.

3. Mandatoriness. Beyond such spatial features, other work highlights the fact that venues can only affect segregation and integration patterns to the extent that they impose an obligation on individuals to visit them–and correlatively that individuals feel committed to do so. A venue cannot reliably anchor interaction in a specific location if there is no requirement to attend it and nobody feels at least somewhat compelled to visit it. For this reason, some authors have featured religious venues as prime examples of how place-based segregation is produced. For example, in their study of Jewish communities, Harold and Fong note that synagogues produce local clustering by demanding regular attendance at a specific location, in virtue of being charged with a special symbolic significance capable of generating sustained commitment [66]. They point out that while this is a particularly pressing matter for orthodox communities, it remains relevant in residential choices for non-orthodox Jews as well, depending on their level of commitment to regular attendance. Similarly, Martinez's (2017) ethnographic study of Italian American Catholic parish members in a predominantly Latino neighbourhood shows how the shared commitment to attend the local church helped to sustain a relatively mixed group of parishioners, even as the church itself became a locus for contestation about community identity. While other examples might be adduced, these serve to highlight the theoretical importance of incorporating into our framework the general mechanism whereby obligation to attend a venue helps to generate place-based patterns of integration and segregation. They also indicate that

"mandatoriness" has two sides: on the side of the venue is its degree of obligatoriness; on the side of the agents is their degree of commitment. Therefore, our simulation framework allows the researcher to examine variations in both venues' "obligatoriness"–how strongly they impose an attendance requirement–and agents' commitment–how likely they are to visit a venue within their catchment area.

4. Openness. Other research points toward an additional and related crucial mechanism through which venues affect urban residential patterns: their openness. By excluding or including particular groups, venues change the expected diversity of local encounters. Mayo [67], points out how, historically, the presence of country clubs helped to consolidate elite neighborhoods by both reinforcing movement to the suburbs and through restrictions on access based on race, income, and status. Jim Crow laws enacted segregation by creating separate schools, hospitals, and restaurants for white and Black Americans [68]. Relatively exclusive venues can also support integration by creating spaces where the in-group can interact, thereby making integration in the wider area more acceptable. Graziul (2016) shows this for the case of some African-American churches. Correlatively, agents' willingness to accept or transcend expectations about who belongs in what venues affects the composition of individuals who attend and interact within a given venue, with more "adventurous" persons being likely to visit more exclusive venues of other groups, while the less adventurous will only visit their own venues and especially inclusive venues of other groups. As an example, consider early-wave gentrifiers such as artists and academics who are attracted to a neighborhood when it still has its traditional composition (Mathews, 2010). The example also illustrates how "adventurousness" and "tolerance" differ–an adventurous yet somewhat intolerant gentrifier might be more likely to enter into relatively exclusive spaces of other groups and find the resulting interactions uncomfortable. To capture "openness," our simulation framework includes both venues' exclusivity–the degree to which a venue is restricted to a single group– and agents' adventurousness—the degree to which an agent is willing to enter into an available venue that is not geared toward their group.

## Methodology

This section describes our methodology, in four parts. First, we characterize the parameters in our simulation framework; second, we define the simulation rules; third, we describe the analytics we use to evaluate simulation results; and fourth, we overview the simulation experiments we conducted for this study.

### Simulation parameters

The parameters in our simulation framework build upon Schelling's (1971) original model and the subsequent literature discussed above, while adding additional parameters to capture the theoretically salient dimensions of venues we have articulated. Parameter definitions adopted from the baseline Schelling model include:

- Simulation World

  - World Size, Shape

    - In our model, the simulation world is a square of 50 by 50 cells.

  - Distance

    - We use Moore measures of distance (see Fig 1).

- Population Density

  - The proportion (0 to 100) of cells that are occupied by individual agents during the simulation run.

  - Agents are not added or removed during the running of a simulation, hence this number remains fixed even while the locations of individuals and their overall distribution will change.

- Agents

  - Location

    - Individuals occupy a single cell in the simulation space at any one time, and only one individual can occupy each cell.

    - Each individual is randomly assigned a location at the beginning of a simulation run.

    - An individual's location can change during a given time step if they are unsatisfied with the makeup of their current neighborhood.

  - Group

    - Individuals are evenly distributed to one of a number of groups at the beginning of the simulation run. In the simulations discussed in this paper, we only consider cases with two groups.

    - Group membership does not change over the course of a given simulation run.

  - Tolerance/Intolerance

    - A value representing their degree of tolerance or intolerance to members of other groups.

    - In a given simulation run, all individuals have the same tolerance value and it is fixed during the course of the simulation.

    - Tolerance values are measured from 0 to 1. 0 represents entirely tolerant individuals–content with being the only member of their group in a neighborhood. 1 represents individuals who are completely intolerant to the presence of any members of a different group in their neighborhood.

  - Neighborhood Distance

    - The distance in cells around an agent's location that they consider to be the extent of their neighborhood.

    - In a given simulation run, all individuals have the same neighborhood distance value, and it is fixed during the course of the simulation.

    - This value can be set between 0 (individuals have no sense of neighborhood) up to the simulation world size (every cell is part of their neighborhood).

To this baseline set of parameters from traditional Schelling models we add parameters to capture the dimensions of venues discussed above.

- Physical features of venues

  - Location

- Venues occupy single cells in the simulation space.

- Their location is set by the experimenter and fixed during the simulation run.

- Cells occupied by venues cannot also be occupied by individuals.

- Number

  - The number of venues is set by the experimenter and fixed during the duration of the simulation run.

- Venue catchment area

  - The catchment area of a venue is the set of surrounding cells from which an individual could travel to attend the venue.

  - Catchment area is a function of agents' "Venue travel distance".

  - "Venue travel distance" is defined as the distance in cells that individuals are willing to travel from their own location in order to visit a venue during a particular time step.

  - In a given simulation run, all individuals have the same venue travel distance value and it is fixed during the course of the simulation.

  - This value can be set between 0 (individuals will not travel to any venues) up to the simulation world size (individuals could potentially travel to any venues in the simulation world).

- Venue group

  - Each venue is affiliated with one group. These affiliations are set by the experimenter as part of the simulation setup.

  - The group affiliation of venues does not change over the course of a given simulation run, even if the variety of individuals from different groups who visit it might change significantly.

- Openness

  - Venue exclusivity

    - Exclusivity represents the degree to which a venue is or is not open to visitors from other groups attending.

    - Exclusivity values are fixed and do not change over the course of a simulation run.

    - Exclusivity values are set between 0 (an entirely open venue) and 1, a venue which can only be visited by individuals with the same group affiliation. For the majority of our simulations, venue exclusivity is set to 1.

    - In our simulations, exclusivity values are set at the level of venue groups. For example, all venues of Group 1 will have the same exclusivity, but this could be a different value from all of the venues of Group 2.

    - During each time step, "exclusivity" interacts with agents' "adventurousness."

  - Agent adventurousness

    - Adventurousness represents an agent's willingness to visit venues that are affiliated with a different group.

- Each individual is assigned their own adventurousness value, based on a random-uniform distribution between 0 and 1. 0 represents an individual who will only ever attend venues of their own group, and 1 represents an individual who is completely open to visiting venues of other groups. This value is fixed for each individual over the course of a given simulation run.

- Adventurousness is a distinct parameter from tolerance/intolerance. Hence, it is possible for an individual in the simulation to be highly adventurousness but also intolerant.

- During each time step, adventurousness interacts with the "Exclusivity" parameter. The adventurousness parameter only matters when "Exclusivity" is less than 1 or greater than 0. As illustrated in rule 2.A.ii below, it is in these situations that it is necessary to determine which agents visit out-group venues.

- Mandatoriness

  - Venue obligatoriness

    - Obligatoriness represents the degree to which individuals feel compelled to visit a venue.

    - Obligatoriness values are fixed and do not change over the course of the simulation run.

    - Obligatoriness values are set between 0 (a venue which no agent will be required to attend) and 1 (a venue that is mandatory for individuals).

    - In the simulations reported on in this paper, obligatoriness is always set to 1, meaning that all individuals who are otherwise able to attend a venue (based on travel distance, group affiliation, and exclusivity) will attend during each time step.

    - We include a description of the parameter here to indicate the full scope of the model we developed and to suggest a direction of further research.

    - During each time step, "obligatoriness" interacts with the agents' "commitment."

  - Agent commitment

    - Commitment represents an agent's internal sense of obligation to visit venues.

    - Each individual is assigned their own commitment value, based on a random distribution between 0 and 1. 0 represents an individual who is entirely indifferent to visiting venues and 1 represents an individual who must visit all venues available to them. This value is fixed for each individual over the course of a given simulation run.

    - During each time step, commitment interacts with the "obligatoriness" parameter.

## Simulation rules

The rules of our simulation model describe how the above parameters can relate to one another. They again build on rules from traditional Schelling models and then add additional rules to identify potential venue effects. Accordingly, during each time step, the simulation model proceeds according to the following rules:

1. The whole population **P** of individuals are listed in random order.

2. For each individual **A** in **P**

A. Evaluate its relationship to each venue **V**. **A** will visit **V** if:

  i. The distance from **A** to **V** is less than **A**'s **Travel Distance**

  ii. And **Openness**: if **A** and **V** are of the same group, or **A** is **Adventurous** enough to visit **V** [$Group_A = Group_v$ or $Adventurousness_A > Exclusivity_v$]

  iii. And **Manditoriness**: if **A** is sufficiently **Committed** to visit **V** [$Commitment_A > 1 - Obligatoriness_v$]

  - Note: There is no limit to the number of venues that **A** can visit in a single time step, given that all of these conditions are met. This corresponds to a single time step in our models representing something like a day, in which an agent would normally have the opportunity to visit multiple venues. Opening up this simplifying assumption to more complex rules determining the number and order of visits would be an interesting extension of our model.

3. For each individual **A** in **P**

  A. Determine whether A is content with its current location:

   i. Count $N_{Same}$, the number of individuals from the same group within **A**'s **Neighborhood Distance**

   ii. Count $N_{Total}$, the total number of individuals within **A**'s **Neighborhood Distance**

   iii. Count $V_{Same}$ the number of individuals from the same group who attended one or more of the same venues as **A**

   iv. Count $V_{Total}$ the total number of individuals who attended one or more of the same venues as **A**

  B. Calculate the fraction of individuals of the same group **A** has encountered to the total number of individuals encountered in this time step and compare this to their Tolerance level: A is therefore unhappy with its location if $(N_{Same} + V_{Same}) / (N_{Total} + V_{Total}) < Tolerance_A$.

   - Note: Since $V_{Same}$ and $V_{Total}$ are only changed when A visits a given venue, the presence of the venue itself in a neighborhood does not influence agents' contentment, nor are they affected by the process of others traveling into or out of the venue. This is a simplifying assumption in the present study that could be worth opening up in future work.

  C. If **A** is unhappy, try to move **A**:

   i. Construct a list **C** of all vacant cells in the simulation world (not occupied by either another individual or a venue).

   ii. For each Cell in **C**, determine whether it is a viable location for **A** based on 3.A.i– 3.A.v

   iii. If none of **C** are viable

     1. **A** remains at its current location

   iv. Else

     1. Randomly select one viable location from **C** (see *Benenson and Hatna 2009 and Fossett and Dietrich 2009*)

2. Move **A** to the new location

Fig 2 visually illustrates how key aspects of these rules operate, while Fig 3 shows how they are sequenced in a single timestep

## Simulation analytics

To evaluate our simulation experiments, we utilize a number of analytics, again building on past literature. Rather than cherry-picking particular results, we use a set of analytics and visualizations to explore the variations in outcome that arise from different combinations of input parameters. For each simulation study, we explore ranges of values for two parameters (typically the agents' Tolerance/Intolerance and Venue Travel Distance). Each parameter is stepped through 20 intervals from a minimum to maximum value, generating 400 combinations. Given the random initial distribution of individuals in our simulation runs, we determine the stability of our results for each of these combinations by running them through 10 uniquely random initial conditions. During the development of the simulation model, and given the relatively low variability in outcomes for most combinations in the parameter spaces, we found 10 runs to be an acceptable balance between constraints such as overall runtime and data storage, and the goal of evaluating the simulation's outcomes. Each simulation is run for 20 time steps, by which point individuals have either found appropriate locations or are locked in place with no prospects of finding a location that will satisfy their goal. As our simulation rules indicate, a great deal can take place in a single time step. In none of the cases evaluated for this paper were there still agents willing and able to move by the twentieth time step.

**Concordance.** Concordance is the measure we use throughout our simulations to determine the relative degree of integration and segregation (see Fig 4). This measure examines each individual agent in the population, and determines what proportion of its immediate neighbors (Moore-1) are members of the same group. We define this as the concordance of the individual. To measure overall segregation or integration of the simulation state, we take

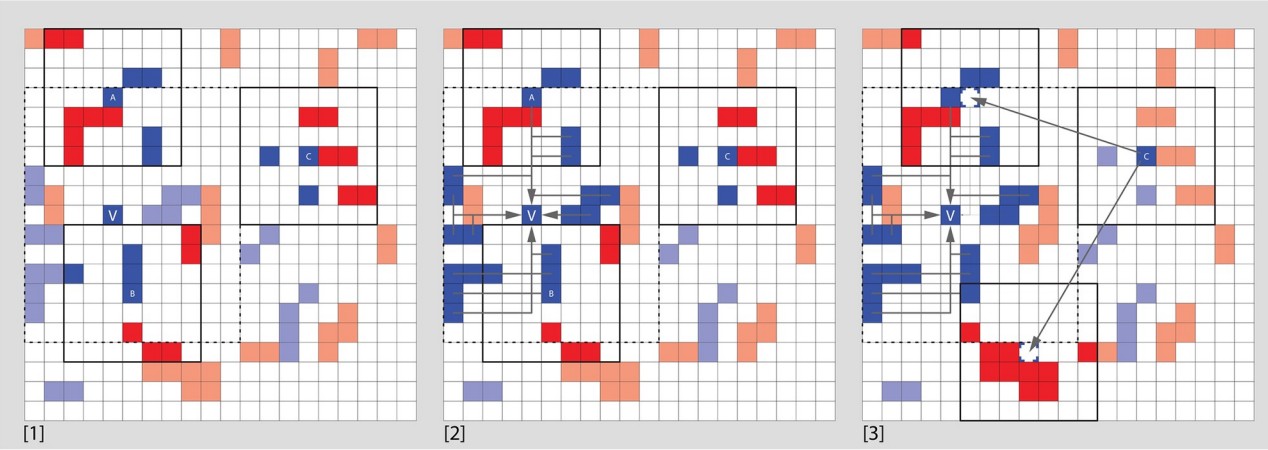

**Fig 2.** [1] A set of individuals and a blue venue. Each individual has its own surroundings and the venue has a catchment area that is determined by the maximum travel distance. [2] If this venue is completely obligatory and exclusive, every blue individual within the catchment will visit during each time-step. Blue individuals beyond the catchment area (such as 'C') are unable to visit. Evaluating their positions, 'A' has 4 like-neighbors of 11, but also 18 like-neighbors added from the venue, yielding a score of 22 of 29. 'B' has 3 like of 8, but also 18 from the venue, resulting in a score of 21 of 26. 'C' has 2 like-neighbors of 8, with no contribution from the venue because it is unable to visit. [3] If 'C' moves this turn, it will consider destinations within the catchment distance of the venue and also beyond in a random order. In evaluating the former, such as location '1', it will consider the influence of individuals from the venue, making it a very good destination, while '2' which is just beyond the venue catchment will not benefit from the additional count, and will hence not be a likely choice.

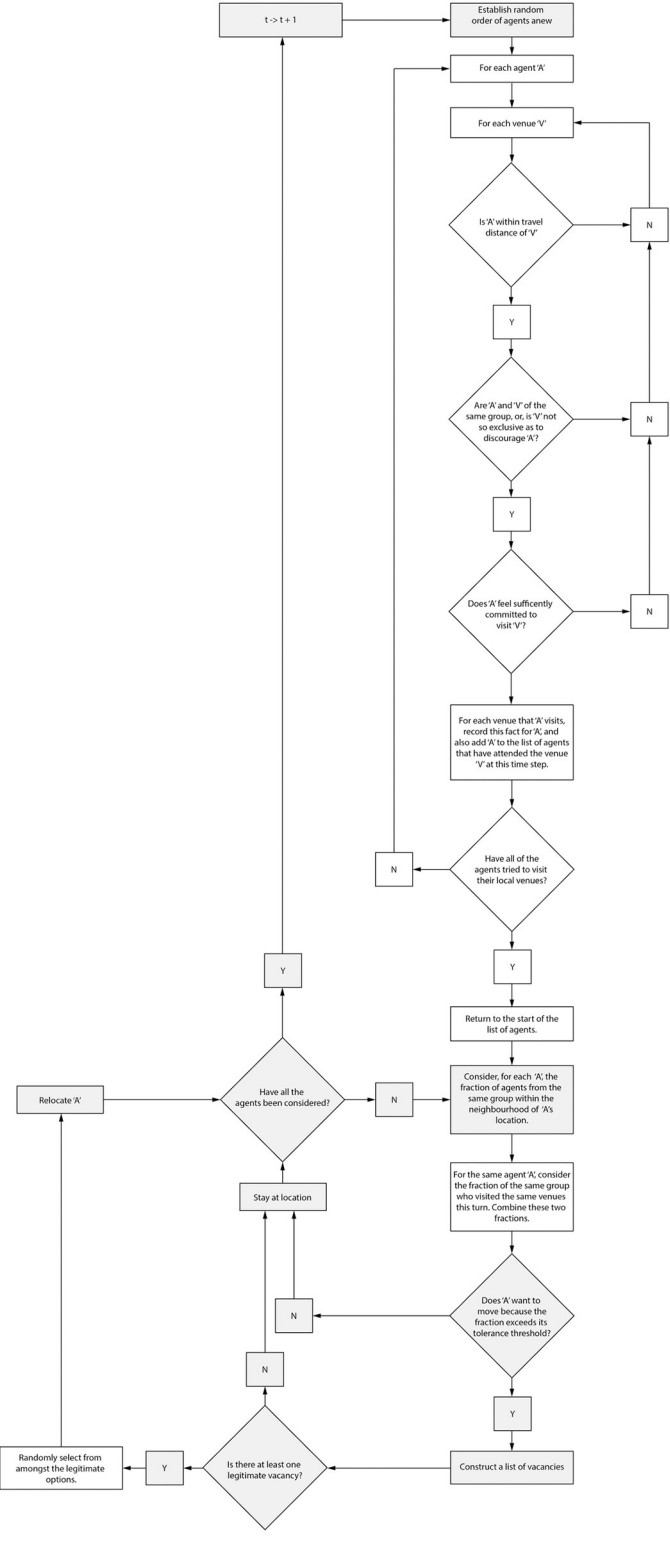

**Fig 3. Flowchart of a single timestep in the simulation model.** Chart elements shaded in grey are typical of Shelling implementations while our own additions have a white background.

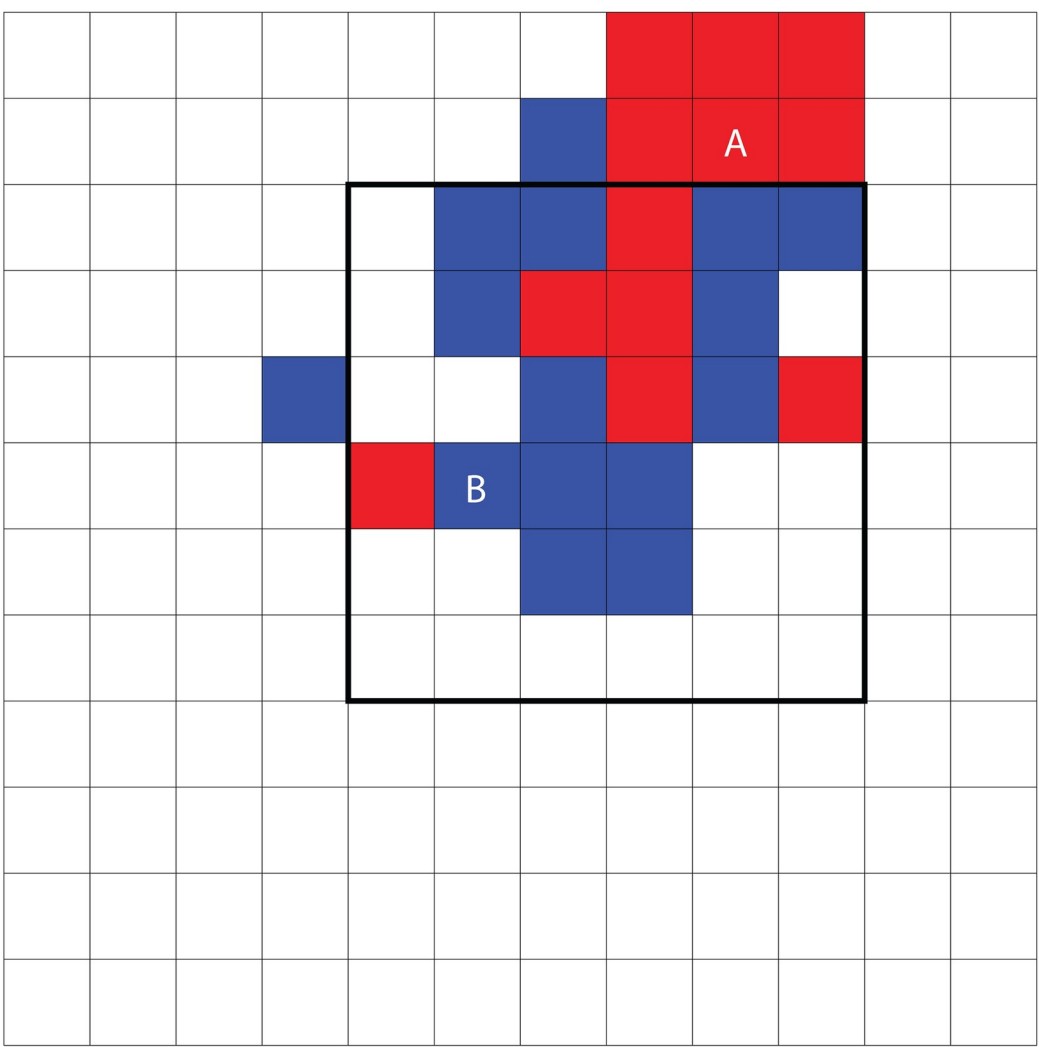

**Fig 4. Illustration of concordance and predominance.** We can measure the concordance of agents A and B by counting what proportion of their immediate neighbors are members of the same group. In the case of A, we have 6 of the same group and 2 of the opposite group, making a total of 8. The concordance of A is hence 6/8 or 0.75. In the case of B, we have three of the same group, and 1 of the opposite group, for a total of 4. The concordance of B is hence 3/4 or, in other words, also 0.75. When it comes to measuring predominance, some area of the simulation world must be identified where the count will take place. In this case, an outlined area is indicated in bold. Predominance is determined by first counting the number of agents in the area of each group. Here, we have 6 red and 13 blue. The predominance in this case is therefore, (13–6)/19, or 0.37.

the average of all these concordance values. An initially shuffled simulation world, in which each individual is just as likely to have a neighbor of the same group or the opposite would hence have a Concordance of approximately 0.5. On the other hand, a completely segregated simulation world, in which empty cells acted as a buffer between the two groups, would have a Concordance of 1 since each individual would only have immediate neighbors of their own group. Schelling himself makes use of the same measure in his 1971 paper (p. 156), describing it as one measure of "segregation or concentration or clustering or sorting", the "average proportion of neighbors of like or opposite color". For the sake of brevity, we provide this measure with a name. Laurie & Jaggie produce their own variation on a demographic dissimilarity index, the "ensemble averaged, von Neumann segregation coefficient at equilibrium". Hatna

and Benenson, throughout their papers make use of the Moran's I index of spatial association, and other more specific analytics based on their particular objectives. Our measure of Concordance has the double merit of being extremely straightforward and in line with Schelling's original thinking. Put another way, the simplicity of Concordance is an appropriate match to the simplicity of the simulation model itself.

**Predominance.** Since our studies explore not only the clustering of individuals of different groups, but also the specific form and location of these clusters in relation to venues, Concordance is not always sufficient for understanding differences in the simulation outcomes. We create a novel metric, Predominance (see Fig 4), as a way of measuring the extent to which one or another of the groups has taken over a particular area of the simulation world. Given some subset of the cells in the simulation (for instance, those within the catchment area of a venue, or those at the center of the simulated world), we determine Predominance by counting the fraction of individuals in this area who belong to the group which is most highly represented there. In other words, it is a measure of the size of the majority held by whichever group holds the majority. Measured on a 0 to 1 scale, 0 represents a situation in which neither group predominates (equal numbers of individuals from both groups are present) whereas 1 represents a situation in which the area in question is exclusively occupied by one of the groups.

**Settings and distribution.** For each simulation study, we begin by listing the two simulation parameters that will be allowed to vary between a given minimum and maximum value (see Fig 5). We also indicate the simulation setup in terms of the locations and group affiliations of any venues. Full simulation settings for the study are contained in the Supplementary Material.

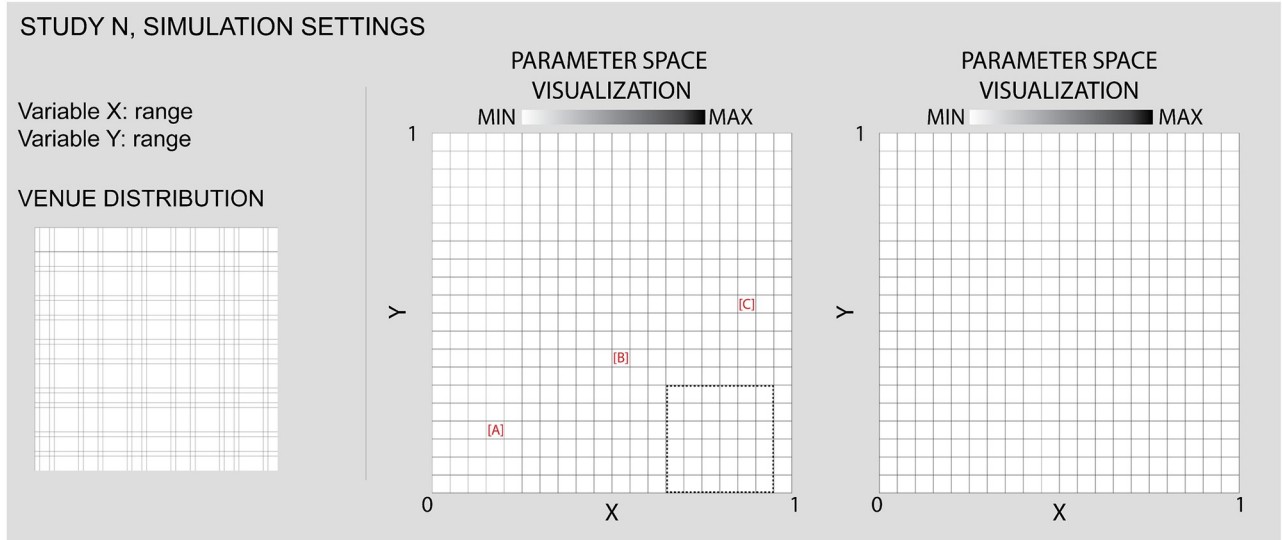

**Fig 5. In reporting our study results, we follow a consistent layout and utilize a repeated set of analytics, illustrated as an empty framework here.** In the top left, specific simulation parameters that vary in the study are listed. These variables become the X and Y axis of subsequent charts. Complete simulation settings are available in the S1 Appendix. Below the parameter settings, we illustrate the locations of venues for this study. To the right of these "input" settings, we illustrate parameter space results for the study. These charts show how values of Concordance, Predominance, and volatility (in terms of Standard Deviation) change with different values of the input parameters, where grey-scale values indicate average values across multiple runs with randomized initial distributions of agents. Dashed shapes within the parameter space indicate that a subsequent chart will focus on a smaller region of the full parameter range, while red letters in square brackets connect with specific images of simulation results illustrated in a companion figure (in this case, in Fig 6 below).

**Parameter space visualizations.** Since we are interested in the outcomes of the simulation not for a single run but across a range of different inputs, we take the input parameters which have been selected to vary and plot them along an xy chart in order to see what patterns emerge between variations in the inputs and simulation results (Fig 5). In these visualizations, individual cells are colored according to the average value of one of our analytics (e.g. Concordance or Predominance) across ten runs of the simulation. These visualizations allow us to determine how different input parameters relate to one another and where 'tipping point' behaviors occur. Furthermore, they provide a context for locating the particular simulation runs that we draw specific attention to with Images and Videos of the simulation worlds.

**Standard deviation.** An important question in the design of simulations generally is their sensitivity to initial conditions. This is a consideration for Schelling simulations and in our model specifically, given that we rely on random initial positions for the population and random distributions of "adventurousness" and "commitment." In order to measure the relative volatility or stability of our simulation results, we generate secondary parameter space visualizations (Fig 5). For each combination of input parameter values, these represent the Standard Deviation in Concordance values across the ten runs of the simulation, as a grey-scaled square. In general, we find the results to be highly stable across different random initial conditions, with volatility increasing only in some areas of the parameter space where tipping points are evident.

**Images and videos of runs.** A major part of our study results are images drawn from the final configurations of simulation worlds after they have been run through twenty time steps (Fig 6). These images provide the basis for our discussion of the different mechanisms at work in the simulation model and, when linked back to the parameter space visualizations (by numerical indicators) allow us to draw conclusions about the different outcomes brought about through changes to the input parameters. Lastly, since some of the patterns we observe are best understood through their sequential development over multiple time steps, we include links within the paper to videos of specific simulation runs in order to illustrate important features of the model's behavior.

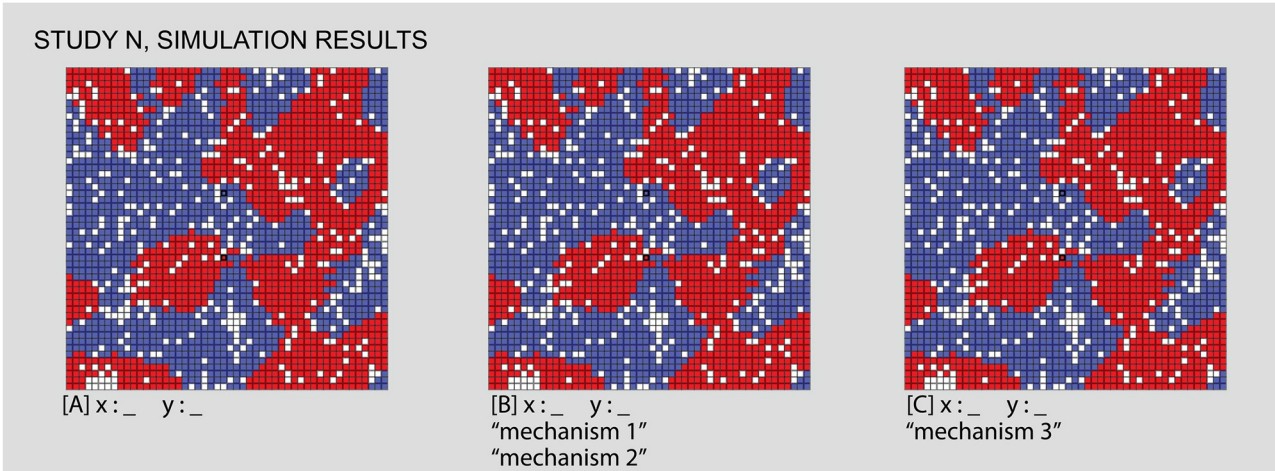

**Fig 6. Here we illustrate actual resulting configurations of the simulation world for different input parameter values.** These result images are keyed to the parameter space illustrations (introduced in Fig 5 above) through the letter system in square brackets to the bottom left of each. Beside this number, x and y values indicate the exact values for the variable input parameters that yielded the outcome. Finally, mechanisms that are catalogued and described as part of our results are also indicated here for the sake of quick reference.

## Simulation experiments

In this paper, we describe three studies designed to examine the impact of venues on segregation. Our studies are designed to investigate how the presence of venues, their properties, and their geography affect the baseline patterns from traditional Schelling models. Study 1 introduces venues into a Schelling simulation, testing simple layouts involving a pair of centrally located but opposed venues. By evaluating these layouts in terms of a range of intolerance and travel distance values, we identify key thresholds in the parameter spaces, the stability or volatility of the outcomes across multiple runs, and visual patterns of agglomeration. These basic configurations allow us to see the influence that venues can have on Schelling simulation outcomes even in a very simplified and minimalistic setting.

In Study 2, we vary venue geography so that we can begin to understand how initial venue geography affects segregation dynamics. In the urban world, a venue's location is often a function of historical processes that are exogenous to the actors and can decisively affect subsequent patters. These are akin to "second nature" geographical conditions identified by Krugman [69]. Venue geography, like the presence of a hill or valley, is an initial condition that the emergent system must adjust to. Study 2 simulates two idealized but recognizable patterns inspired by empirical findings in North America: in study 2.1, a concentric circle (Burgess, 2012; Schwirian, 2007); in Study 2.2, a core-periphery pattern (Florida and Adler, 2018; Qi et al. 2004). These studies investigate how contingent neighborhood mixing is on familiar patterns of venue location.

Study 3 considers the role of venue-level planning in shaping neighborhood dynamics in a simple bifurcated arrangement of venues. This arrangement allows us to evaluate whether varying the Exclusivity parameter could impact segregation in a linearly divided neighborhood formation with multiple venues. Large scale social change often emerges from the micro to the macro, either via uncoordinated actions or planning. The idea that a community would uniformly change their attitudes toward another group (i.e simultaneously lower their intolerance variable) strains credulity. On the other hand, action at the venue level can and does often eventually lead to higher order effects. Here we consider how the reformatting of a venue to be less exclusive affects the neighborhood mix. More specifically, by varying venues' Exclusivity, while holding constant their spatial arrangement, Study 3 demonstrates that the social rules governing venue attendance alter segregation patterns. In turn, by varying the Exclusivity value, we are able to observe situations in which agents' Adventurouness can lead to distinctive patterns of segregation or integration around venues. In this study, we limit the variations in Exclusivity to the blue venues, while varying the intolerance level of both groups, in order to observe the introduction of some red agents into blue venues. This allows us to examine effects produced by allowing interactions between individuals of different groups within venues.

Together, these studies demonstrate the utility of embedding the Schelling model in a simple urban environment. In the course of testing the implications of varying the parameters featured in these three studies, we discovered several recurrent mechanisms that generate overall patterns of segregation, such as "locking in, "bridging,", "bootstrapping," and "abandonment." We discuss and define these as they arise in each study, and collate them in the Conclusion.

## Findings

We now report key results from the three studies described above. Our discussion of findings centers on the recognizable urban patterns that are generated when venues are overlaid on familiar Schelling processes and the mechanisms that appear to produce these patterns. Eight generative mechanisms emerge in the course of the analysis: in Study 1, "locking in,"

"bootstrapping," and "action at a distance"; in Study 2, "bridging," and "abandonment"; and in Study 3, "cascading," "evacuation," and "co-optation."

## Study 1: Access to venues changes the tendency of agents to segregate

**Study 1.1. The introduction of two opposing venues changes the outcome of the original Schelling model.** Study 1 demonstrates that the introduction of venues can change the outcome of the original Schelling model, in particular by generating place-based patterns of segregation that are characteristic of urban life.

As Figs 7 and 8 show, this study positions two venues (each associated to one of the two groups) just north and south of the world's center. These venues are entirely exclusive and obligatory. We run the simulation while varying the individuals' intolerance levels from 0 to 1 and their maximum travel distance from 0 to 50 (i.e. the size of the simulation world). The center and right panels of Fig 7 represent the simulation's parameter space, and the views in Fig 8 show the simulation outcomes for certain combinations of input parameters, as described above. Numerical labels in the parameter space again correspond to the simulation views, linking them together and allowing us to refer to outcomes of particular interest.

Figs 7 and 8A: *Classic Schelling segregation and the role of "locked in" agents.* As a baseline for the introduction of venues and an introduction to interpreting results from our simulations, we first demonstrate that our model can reproduce Schelling's classic results. We also identify a mechanism by which mixed neighborhoods can arise when agents become "locked in" to their current locations.

The classic Schelling model occurs on the bottom row of the parameter space, where the maximum venue travel distance is 0. Since a value of 0 means that venues cannot be visited, this effectively removes any venue effects from the model. Looking at this row of squares, we see light squares on the left side, where individuals have such a low intolerance that they are satisfied with their randomly mixed initial position. Moving to the right, the classic Schelling tipping point is represented by the increasingly dark squares starting from an intolerance of

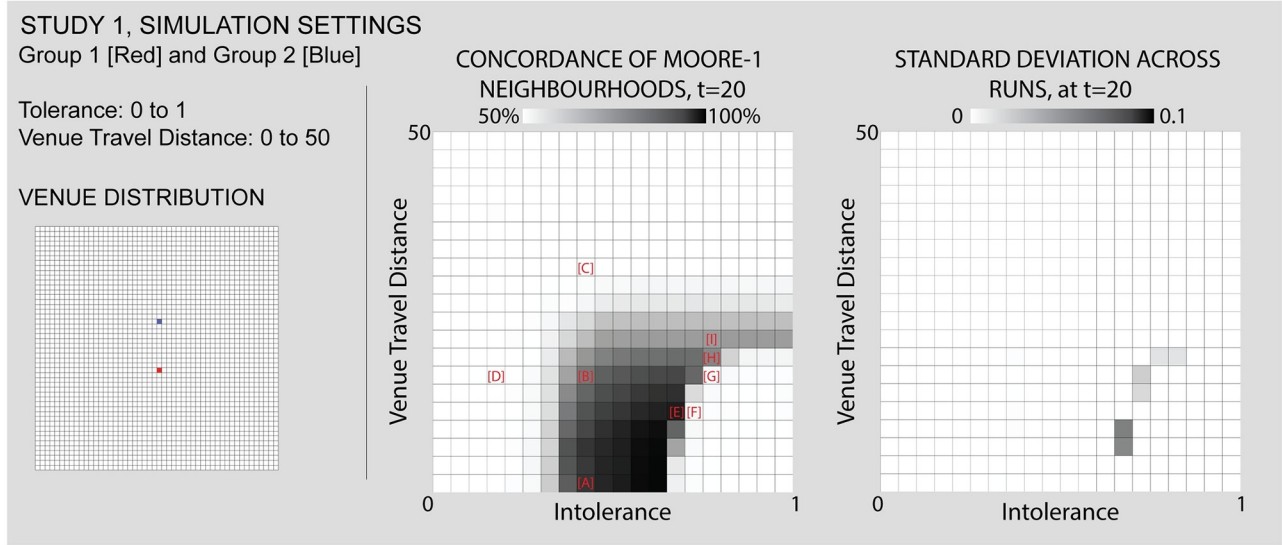

**Fig 7. Study 1 simulation settings and resulting parameter space.** For additional simulation settings, see the S1 Appendix. Letters in square brackets within the parameter space refer to specific simulation runs that are illustrated in Fig 8.

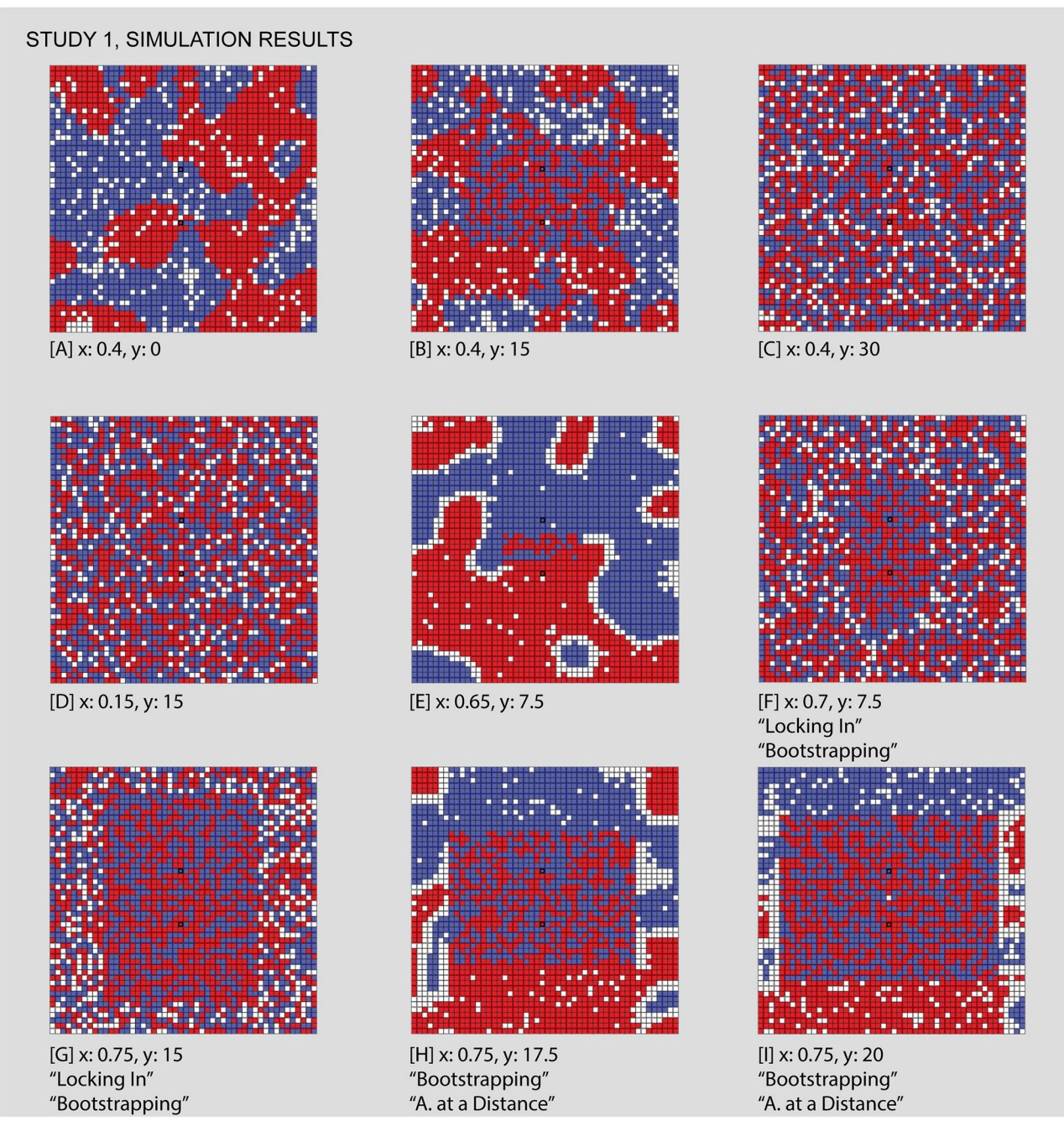

**Fig 8. Results sampled from Study 1, comparing different values for individuals' intolerance (x) and the maximum travel distance to venues (y).** In the first view, [A], we see classic Schelling-style segregation, [B,C] the influence of venues as travel distance increases, [D] an area of low intolerance, [F,G] "locked-in" situations that in [E,H,I] are "unlocked" by the influence of venues, which "bootstrap" new neighborhoods. Each of the letters in square backets can be used to locate these specific results in the larger parameter space by referring back to the red letters in Fig 7.

0.35. As intolerance increases, the concordance of each cell with its neighbors increases, or put otherwise, segregated neighborhoods emerge.

Moving further to the right demonstrates a different mechanism by which mixed areas can arise: "locking in" (see Table 1) "Locking in" occurs when a group of individuals are unsatisfied

**Table 1. Generative processes discovered during the simulation case studies.**

| Mechanism Name | Description | Sim. Case |
|---|---|---|
| Locking in | A group of individuals are unsatisfied with their current locations and yet none of the available cells are suitable to motivate a move. They are stuck in surroundings with higher diversity than they desire until some change in the simulation world "unlocks" the situation by opening up suitable destinations for these individuals. | Study 1 (Figs 7, 8F and 8G) |
| | | Study 2.2 (Figs 11 and 12D) |
| | | Study 3 (Figs 13 and 14E) |
| Bootstrapping | Venues are able to generate highly integrated neighbourhoods, even in cases of highly intolerant populations, by turning all of the individuals of a given group within the catchment into neighbours. Access to the venue with a high number of like-individuals makes the integrated immediate surroundings acceptable in spite of the individuals' intolerance. | Study 1 (Figs 7 and 8F–8I) |
| Action at a Distance | The circulation of individuals caused by venues can have an influence even beyond their catchment area. For example, as individuals leave to move closer to a venue, the cell they leave becomes available for others to occupy, potentially leading to the formation of new neighbourhoods. Especially near the perimeter of a venue's catchment, such movement can manifest as an extension of the venue's neighbourhood. | Study 1 (Figs 7 and 8H and 8I) |
| Abandonment | When a group within the population that is driven by intolerance to a highly concentrated district, and as a result leaves the catchment area of one or more of its venues altogether. Typically such a group will be concentrated around one or more of its other venues. | Study 2.2 (Figs 11 and 12B) |
| Bridging | An area between two venues that are associated with the same group can appear to be coordinated under the influence of the venues, leading to a continuous segregated neighbourhood that connects the two catchment areas. This effect can be observed even if these cells are not within the catchment area of either venue. | Study 2.1 (Figs 9, 10B and 10C) |
| | | Study 2.2 (Figs 11, 12A and 12B) |
| Cascading | A critical mass event involving the departure of individuals from an area. A cascade is a relatively slow departure that begins from the most adventurous individuals of a group visiting a venue belonging to the opposite group and subsequently departing from the area. Their departure leads more and more individuals to leave, eventually impacting even the least adventurous amongst the group. | Study 3 (Figs 13, 14B and 14C) |
| Evacuation | The abrupt departure of individuals from an area surrounding a venue of the same group. The evacuation results from a combination of relatively high intolerance amongst the individuals and the presence of at least a small number of individuals from the other group who are sufficiently adventurous to attend the venue. Their attendance at the venue makes the area intolerable for the other individuals at the venue. | Study 3 (Figs 13, 14F and 14G) |
| Cooptation | If a small number of individuals force the "evacuation" of the other group from their own venue, and the resulting vacancy encourages more individuals of the new group to arrive, these individuals can become the majority within the venue, even though it is associated to the other group that has departed. | Study 3 (Figs 13 and 14H) |

In the course of these simulation experiments, a set of generative processes emerged, which we catalog and describe in Table 1. With this approach, we follow Kaidesoja's (2019) suggestion that theory-development seek to build up an expanding cluster of mechanism-based explanations that may be tested in new domains.

with their current locations and yet none of the available cells are suitable to motivate a move. They are stuck in surroundings with higher diversity than they desire until some change in the simulation world "unlocks" the situation by opening up suitable destinations for these individuals. As intolerance continues to increase, "locking in" becomes more common. Individuals are increasingly unsatisfied with their initial locations, and yet none of the available cells are suitable to motivate a move. While Schelling's insight is that relatively low levels of individual

intolerance can lead to segregation at a macro scale, in these cases the reverse is also true: high levels of individual intolerance lead to macro diversity, albeit among agents who are unhappy with their location. Comparing the two kinds of mixing to the left and right of the first row is hence a matter of accounting for individuals' satisfaction with their mixed neighborhood. The satisfactory diversity in the left side is qualitatively different from the undesired diversity on the right.

More than an artefact, this locking behavior reflects real conditions in expensive urban land-markets where housing constraints prevent the production of new residences (Hsieh and Moretti, 2019). It is common for residents to be locked out of satisfying their preferences. More generally housing supply is constrained by zoning practice which artificially restricts the number of available residences in urban settings (Manville et al. 2020).

Figs 7 and 8B–8D: *Venues can generate zones of integration that grow wider as their catchment areas expand and can compensate for relatively high levels of intolerance.* Comparing other areas of the parameter space to the baseline established in Figs 7 and 8A shows how venues impact the simulation outcomes. Consider first the impact of the presence and catchment area of a venue. Reading upwards from Figs 7 and 8A, each square is a lighter shade than the one before, meaning that the outcomes are increasingly mixed. By looking at Figs 7 and 8B, we can see why this is the case: even while the general pattern is towards segregation, the area within travel distance of both venues is highly integrated. In this area, individuals are comfortable with a more diverse neighborhood because they participate in a venue with many individuals of their group (see S2 and S3 Videos for a comparison of the same initial distribution of agents with and without venues). Even with relatively high levels of intolerance, it is only the individuals without access to the venues that feel compelled to segregate. The venue acts as a kind of bait, making the mixed environment more acceptable.

As the maximum travel distance increases, the size of this mixed area also increases. Eventually, as in Figs 7 and 8C, the maximum travel distance encompasses the complete simulation world, at which point the importance of the venues entirely outweighs each agent's local neighborhood. While the area of low intolerance to the left of the parameter space visually resembles Figs 7 and 8C, the two are importantly different. In Figs 7 and 8C, individuals are unsatisfied with their initial random neighborhood, but access to the venue compensates for this fact. While in Figs 7 and 8D, when intolerance levels are very low, the positive impact of the venues is unnecessary because each individual is already satisfied with their neighborhood and hence none feel compelled to move.

Figs 7 and 8E–8I: *Venues can create the conditions for mixed neighborhoods through a mechanism we call "bootstrapping".* "Bootstrapping" occurs when venues are able to generate highly integrated neighborhoods, even in cases of highly intolerant populations, by turning all of the individuals of a given group within the catchment into neighbors. Access to the venue with a high number of like-individuals makes the integrated immediate surroundings acceptable in spite of the individuals' intolerance, thus unlocking these cells as potential destinations. Comparing Figs 7, 8B and 8F illustrates this sort of bootstrapping. In the third area of mixed outcomes in the lower right of the parameter space, agents without venues are 'locked in' to a diverse geography that they are dissatisfied with. As the travel distance increases (Figs 7 and 8F), these unsatisfied agents are drawn towards any available cells within range of the venues. Like the individuals near the center of Figs 7 and 8B, they are content with the mixed surroundings near the center because these cells provide access to many members of the same group at the venue. Unlike in Figs 7 and 8B, the level of intolerance in Figs 7 and 8F leads to significant pressure towards the center, evidenced by the high density (the absence of vacant cells) within range of the venues. Thus, venues seem to bootstrap new mixed neighborhoods of which they become the center points. By leveraging all the like-individuals within travel

distance of the venues as potential neighbors, the nearby empty cells become suitable destinations, drawing agents in from the periphery.

An additional mechanism appears from Figs 7, 8F to 8E and 8G to 8H, where venues "act at a distance" to break the locked-in condition that we observe in earlier runs, supporting the formation of new segregated areas on the periphery. "Action at a distance" refers to situations when the circulation of individuals caused by venues can have an influence even beyond their catchment area (see Table 1). In Figs 7, 8E, 8H and 8I, all of the movement inwards towards the venues has the side effect of opening up satisfactory positions around the periphery. Individuals who are beyond the maximum travel distance from the venues nonetheless succeed in organizing themselves into segregated neighborhoods that satisfy their preferences as a result of other movements caused by venues. For example, at Figs 7 and 8E we can see a typical pattern of Schelling clustering beyond the overlapping areas of the venues. This manifests as a curved threshold in the parameter space, where different combinations of travel distance and intolerance interact with the venues and lead to two possible outcomes. On one side of this threshold Figs 7, 8F and 8G, agents outside of the venue range are still "locked-in", while on the other side Figs 7, 8E, 8H and 8I, the combination of parameters enables these agents to be unlocked and to segregate in conventional Schelling patterns.

Thus, not only do venues produce combinations of integrated and segregated neighborhoods around themselves, they even indirectly influence the decision-making of individuals beyond their catchment areas. These cases generate a highly mixed core that is familiar in many cities, where access to the venues enables individuals to accommodate much more diversity in their immediate surrounding, but also produces a highly segregated periphery. This illustrates how venues create place-based patterns of integration and segregation.

## Study 2: Different spatial distributions of venues yield different neighborhood shapes

Study 2 examines how the spatial distribution of venues affects the patterns of segregation and integration that emerge around them. Study 2.1 examines a radial pattern, while Study 2.2 investigate the effects of a core-periphery arrangement.

**Study 2.1. A central space with privileged access to a surrounding ring of venues creates a "contested core" with highly variable outcomes.** Beginning with the radial model, across multiple iterations we find the central area to exhibit a high degree of sensitivity to the initial random distribution of individuals. This is revealed in the Predominance representation of Fig 9, which counts the extent to which the one group exceeds the other in occupying the central area delineated by the venues. Unlike the tipping of neighborhoods in the abstract grid of Schelling's model, or even the simple distributions described in the first study above, the radial model is made more interesting insofar as the contested core–with its privileged access to the whole ring of venues–represents a distinctive urban area that would be desirable for a group to occupy. Further, space in this model is hierarchical. In Fig 10 we see that the same sorts of conditions which give rise to the "cosmopolitan" diversity of the urban core also can make it into a zone of competition and contestation in which outcomes are uncertain and dependent on the initial distribution of agents.

Figs 9 and 10A: *In the radial configuration, the presence of multiple venues associated with the same group supports expanded segregation through a "Bridging" mechanism.* Bridging occurs when multiple venues of the same group appear to generate a larger surrounding area than would be generated by individual venues acting alone. It is a super-additive segregation effect. This bridging can be seen in S4 Video, where connections between venues begin early in the simulation and are increasingly reinforced by the process of segregation

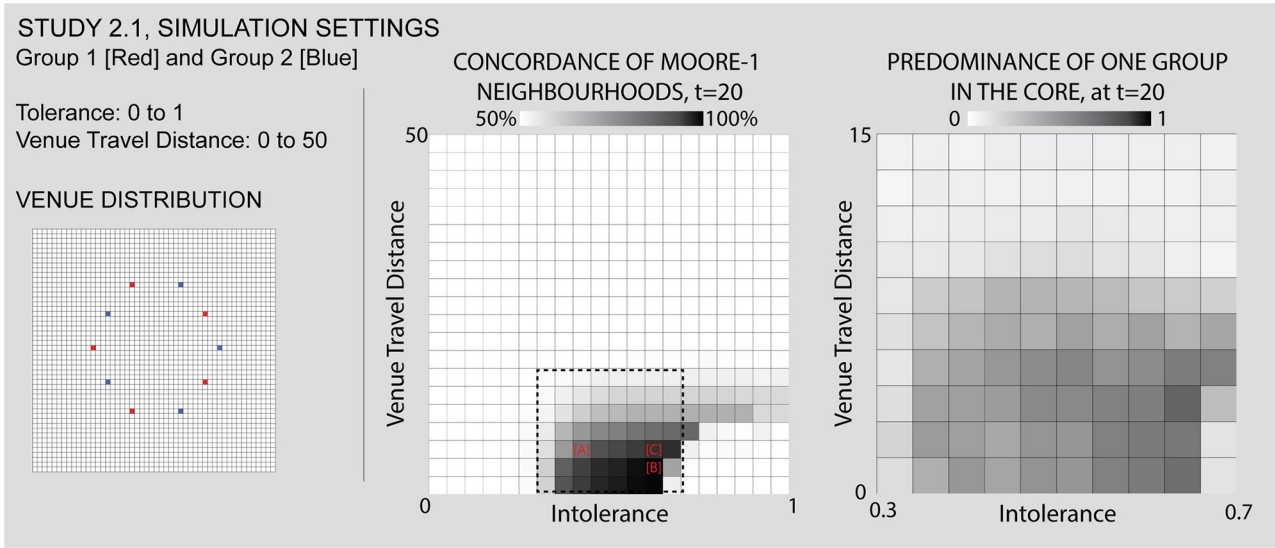

**Fig 9. Study 2.1 simulation settings and resulting parameter space.** For additional simulation settings, see S1 Appendix. Letters in square brackets within the parameter space refer to specific simulation runs that are illustrated in Fig 10.

At Figs 9 and 10B we can see agents "bridging" between some blue and red venues by occupying part of the core, at Figs 9 and 10C we see the blue venues "bridging" directly through and dominating the center area (we see a similarly contested core play out in S5 Video). Even though they yield very similar Concordance values, these two outcomes (Figs 9, 10B and 10C) represent significantly different urban configurations in terms of the predominance of one group or the other within the core area.

*The number of and distance between venues alters their joint impacts.* By increasing the maximum travel distance, we begin to see the overwhelming effect of the venues, where at first Figs 9 and 10B segregation reigns and then Figs 9 and 10C diversity is supported by the expanding

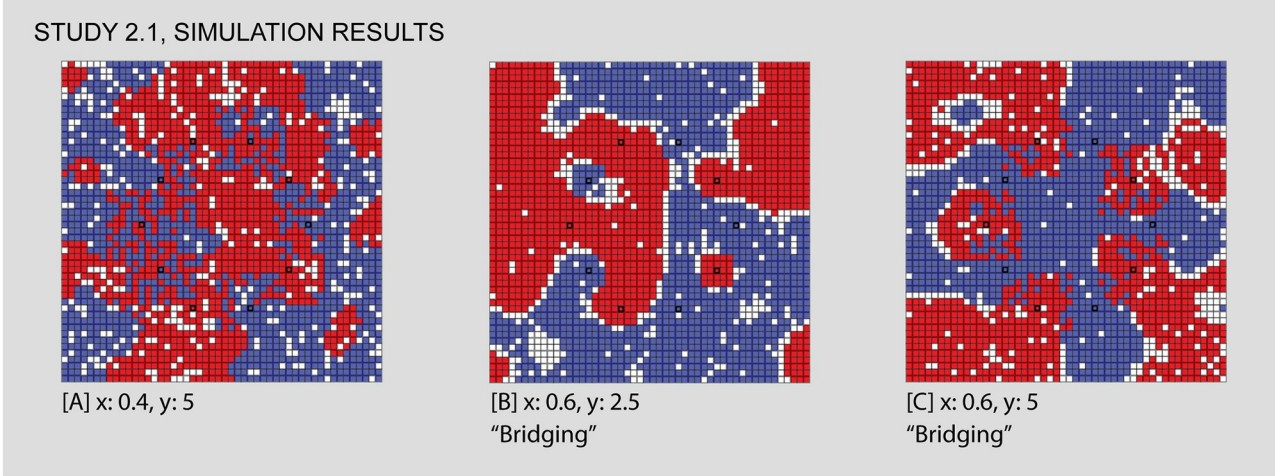

**Fig 10. Some results for the radial distribution.** [A] shows the influence of a radial arrangement of venues on Schelling-style integration, while [B,C] show bridging and take-over situations in the core. Each of the letters in square backets can be used to locate these specific results in the larger parameter space by referring back to the red letters in Fig 9.

circular overlap of catchment areas. Comparing this trajectory of increasing travel distance at high intolerance to Study 1, we can see that the venues break the "locked-in" condition earlier. These differences can be accounted for by the fact that there are more venues and hence the coverage of their catchment areas increases more quickly. Once again, travel range acts as a countervailing force to segregation.

Overall, this study is a strong example of how venues can generate spatial structures beyond their catchments, through a bridging mechanism. Bridged venues create halo effects that increase the likelihood of segregation. Bridging occurs in scenarios with relatively high levels of intolerance but limited travel distances. Even though the catchment areas of these venues do not overlap, they appear to spill into contiguous segregated zones. The venues yield highly segregated surroundings, which continue tipping outwards beyond the catchment. While related to the "action at a distance" from Study 1, where we noted an unlocking of agents, here it is more clearly a coordination of agents between venues, even while they visit neither venue.

**Study 2.2: A core-periphery layout of venues heightens the tendency towards an integrated core and segregated periphery.** Figs 11, 12A and 12B: *Segregation persists in a dense core with low travel distances, sometimes producing "abandoned" areas.* In these two examples of low travel distance, we see the tension between a desire to segregate on the part of individuals and the dense integration of venues. Fig 11 shows settings and Fig 12 results. For example, note in Figs 11 and 12A the diagonal pattern of blue individuals "bridging" between venues, as well as small pockets of red clusters. Towards the upper right of the core area, one red venue holds only a few red agents within a larger cluster of blue. In Figs 11 and 12B and S6 Video, where intolerance is higher, the pattern of venue-driven segregation in the core is more evident, with the exception of one red and one blue venue left of the center.

These runs reveal a process we term "abandonment." Abandonment occurs when a group is driven by intolerance to a highly concentrated district, and as a result leaves the catchment area of one or more of its venues altogether. In Figs 11 and 12B, two venues have been "abandoned" all together, they failed to sustain any individuals from their groups. It is important to bear in mind that since these venues are entirely exclusive, it is not that the other group is

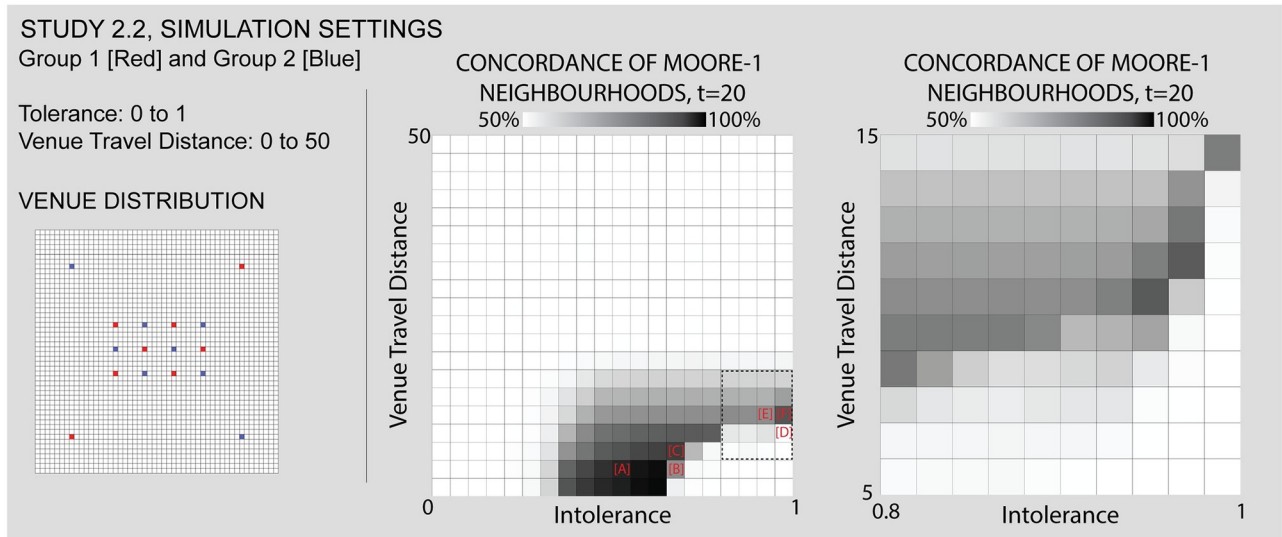

**Fig 11. Study 2.2 simulation settings and resulting parameter space.** For additional simulation settings, see S1 Appendix. Letters in square brackets within the parameter space refer to specific simulation runs that are illustrated in Fig 12.

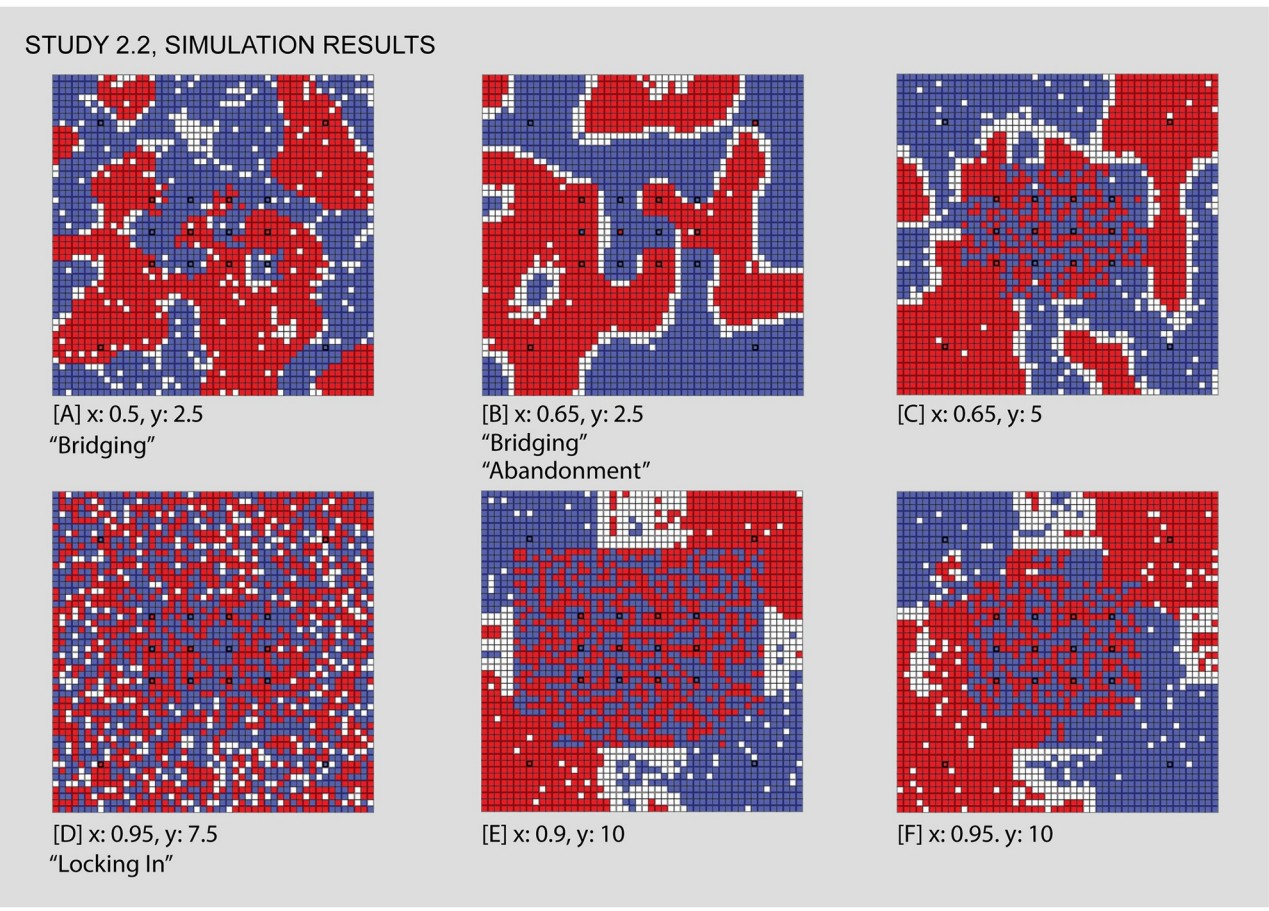

STUDY 2.2, SIMULATION RESULTS

[A] x: 0.5, y: 2.5
"Bridging"

[B] x: 0.65, y: 2.5
"Bridging"
"Abandonment"

[C] x: 0.65, y: 5

[D] x: 0.95, y: 7.5
"Locking In"

[E] x: 0.9, y: 10

[F] x: 0.95. y: 10

**Fig 12. Some results from the core-periphery model.** While study 1 yielded core-periphery patterns, here we test reinforcing this tendency with venues. We note cases such as [A,B] where a fully integrated core does not yet emerge, and [C] where it does. Between [D,E, and F] we note a shift in terms of the relative influence of core and periphery on the overall Concordance. Each of the letters in square backets can be used to locate these specific results in the larger parameter space by referring back to the red letters in Fig 11.

occupying them. Instead these are cases where the positive influence of the venues on its own group is insufficient to support the location. While the distribution of segregated individuals within the core is determined in large part by the venues, we see exceptions in both Figs 11, 12A and 12B.

Figs 11 and 12C *A dense core with greater travel distances produces greater integration.* As the maximum travel distance increases, the core changes from small segregated clusters to a single integrated area. This is in keeping with our findings in Study 1 –that overlapping catchment areas of exclusive and opposite venues support integration. Beyond the core we continue to witness a segregated periphery, with the corner venues beginning to provide coordination to the distribution of individuals.

Figs 11 and 12D–12F *Higher intolerance coupled with larger catchment areas can lead the segregated periphery to predominate.* As travel distance increases, the growing core that we have observed tends to predominate, leading to increasingly integrated results. There are, however, a set of simulations with very high levels of intolerance, in which the segregated periphery pushes back against the integrated core. At the highest levels of intolerance, and with just enough travel distance to avoid "locked-in" outcomes such as Figs 11 and 12D, we can see the tension between solutions where (even with the same maximum travel distance) either the

integrated core predominates as in Figs 11 and 12E and S7 Video, or the segregated periphery does, as in Fig 12F. Fig 14 shows results.

## Study 3—The reformatting of venues leads to surprising and incidental changes in neighborhood geography

We now consider the role of venue-level planning in shaping neighborhood dynamics, examining how varying the exclusivity of venues produces characteristic forms of integration or segregation. Fig 13 shows settings and Fig 14 results.

**Figs 13 and 14A–14D less exclusive venues can sometimes increase segregation by initiating a process we call "cascading".** "Cascading" refers to a critical mass event involving the departure of individuals from an area. A cascade is a relatively slow departure that begins from the most adventurous individuals of a group visiting a venue belonging to the opposite group and subsequently departing from the area. Their departure leads more and more individuals to leave, eventually impacting even the least adventurous amongst the group. This chain reaction makes interactions between agents who do and do not attend venues possible, leading to new types of venue effects that spill out into their surrounding areas. For example, we find that in some situations decreases in a venue's exclusivity can increase segregation, by bringing individuals into interaction who would not otherwise have met, and in turn setting off a chain reaction.

Just as individuals can coexist in an integrated neighborhood with sufficiently low levels of intolerance, so too can they co-occupy venues with certain combinations of tolerance and exclusivity (as illustrated in the light area at the left of the Concordance parameter space in Fig 13). Yet the balance can be quite precarious: depending on how exclusive venues are and how many adventurous agents of the opposite group are nearby, cascading can be initiated due to interactions at the venues. Since exclusivity controls the fraction of individuals from other groups who can attend a venue, as exclusivity increases, those few individuals of another group who are adventurous enough to attend must also be quite tolerant in order not to be repelled from the neighborhood. This relationship appears in the sloping threshold up and towards the

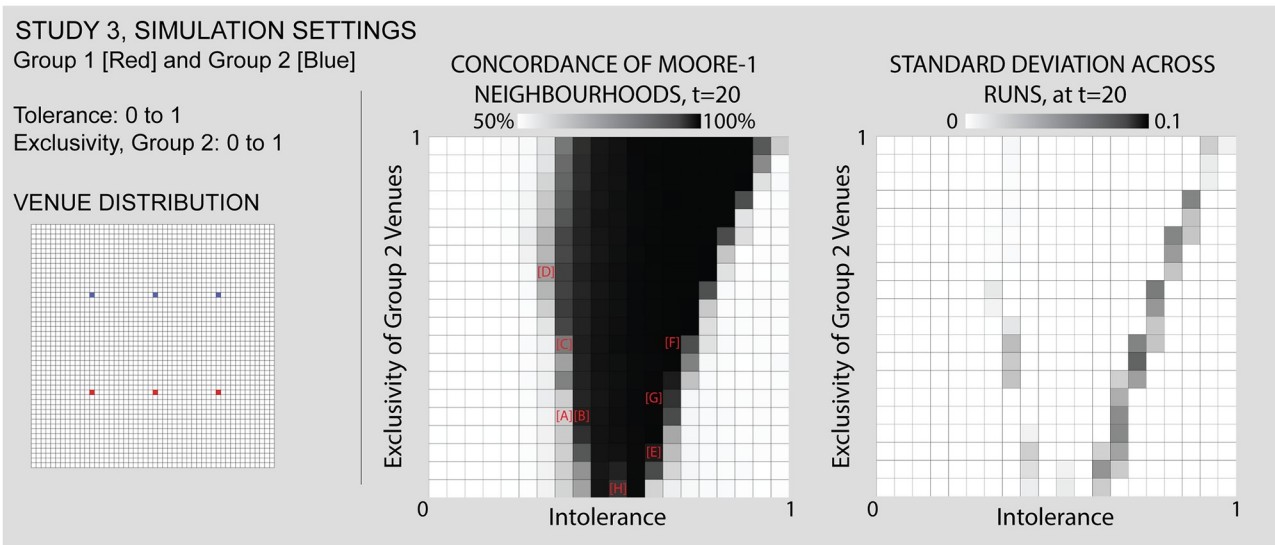

**Fig 13. Study 3 simulation settings and resulting parameter space.** For additional simulation settings, see S1 Appendix. Letters in square brackets within the parameter space refer to specific simulation runs that are illustrated in Fig 14.

STUDY 3, SIMULATION RESULTS

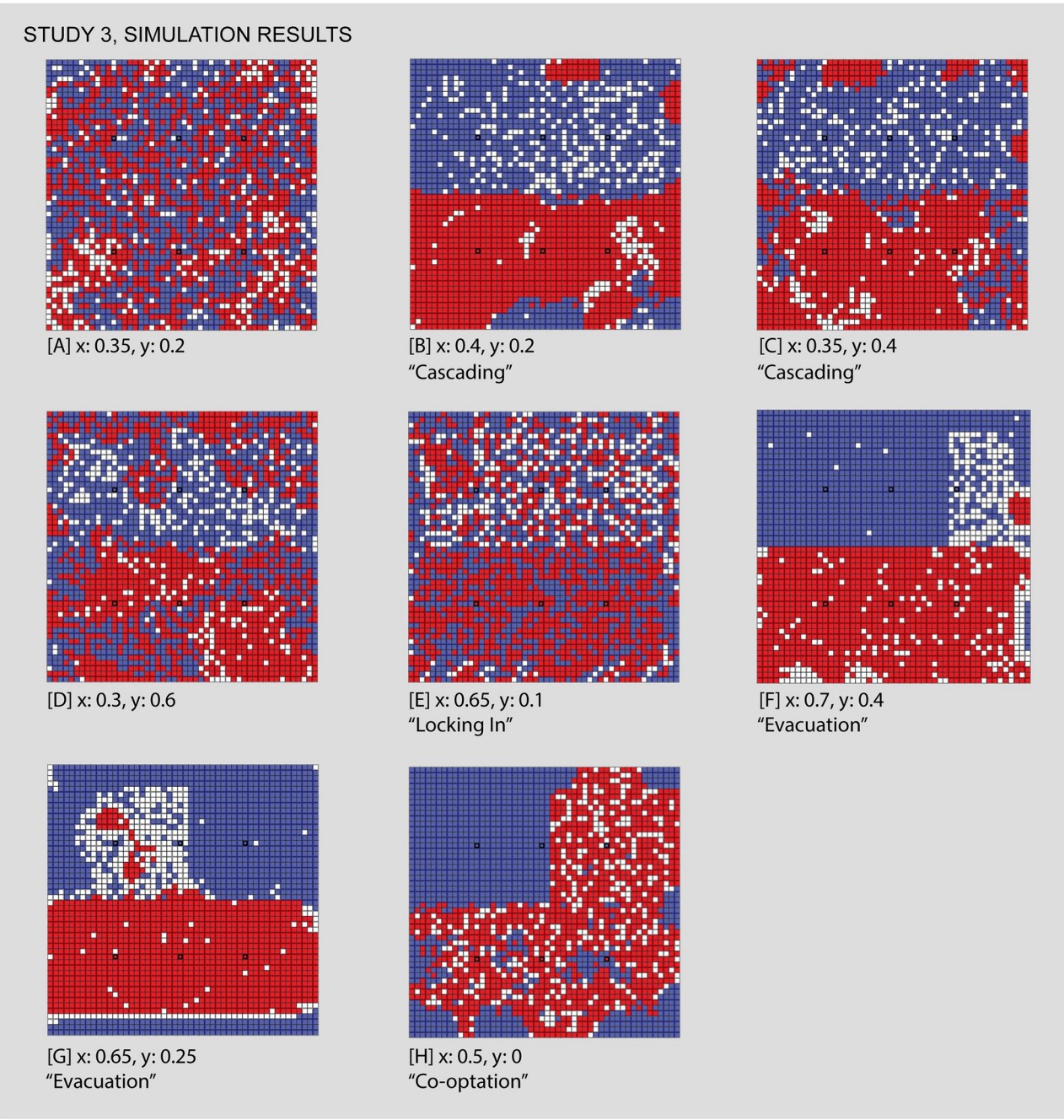

**Fig 14. Some results of the second study, comparing different values for individuals' intolerance and the exclusivity of venues associated with Group 2 (blue).** The focus of this study is the sharing or co-opting of less exclusive venues by another group and the influence this has on the distribution of individuals. In the views above, we see this playing out in different ways: in [A], a sharing of low exclusivity venues amongst individuals of threshold-levels of intolerance, [B,C,D] the other side of the threshold, where the tipping of the venues leads to segregated outcomes, [E] an asymmetrically locked-in condition, and [F,G,H] various levels of neighborhood evacuation and co-opting by another group. Each of the letters in square backets can be used to locate these specific results in the larger parameter space by referring back to the red letters in Fig 13.

left from Figs 13 and 14H to 14D in the parameter space. On one side of this threshold Figs 13 and 14A, individuals are largely integrated, both in the space of the simulation cells and in their attendance of blue venues. The venue is more attractive to the red agents, but the blue agents don't mind having more reds as neighbors because they are more tolerant.

However, as either the level of intolerance Figs 13 and 14B or the exclusivity Fig 14C increases, the situation changes dramatically. With greater intolerance, red agents visiting the blue venues are more unhappy with the large number of blue agents they meet and hence relocate to predominantly red areas. The departure of the more adventurous red agents begins a critical mass event, "cascading" over time and leading eventually to the departure of even those red agents who had not visited blue venues. Without an increase in intolerance, high levels of segregation can also develop when the blue venues are more exclusive. For example, in Figs 13 and 14C, fewer red agents will visit blue venues. Even if relatively tolerant, the adventurous few who attend will decide to leave as a result and initiate the "cascading" departure of all red individuals. Finally, note that when blue venues are yet more exclusive, fewer red agents attend and then leave. The fewer such red agents, the less likely their behavior is to set-off a cascade amongst the others. This relationship is visible in the change of direction around Figs 13 and 14D where the parameter space threshold begins moving upwards towards the right: with fewer red agents repelled by blue venues, intolerance must be higher for their departure to set off a larger cascade. Overall, these runs demonstrate that in some circumstances venues' social rules produce surrounding neighborhoods with seemingly opposite meanings: sometimes exclusive venues can produce more mixed neighborhoods, just as inclusive venues can produce greater surrounding segregation.

**Figs 13 and 14E low exclusivity can "lock in" agents, generating distinctive forms of density around venues.**   In contrast to Figs 13, 14A and 14B where low levels of exclusivity lead to either integrated or highly segregated outcomes depending on the level of intolerance in Figs 13 and 14E we see that low exclusivity can also produce another example of a "locked-in" distribution. Because there are many blue and red agents visiting the blue venues, these venues are unable to attract enough blue agents and the blue agents are not able to "bootstrap" a neighborhood because of the mix in attendees. Blue venues, then, are no longer desirable amenities to blue agents, meaning that no new blue agents will move to fill that region.

We can compare this to the many red agents who are meanwhile drawn to the fully exclusive red venues in spite of the blue agents surrounding these locations. The difference between the effectiveness of the red venues and failure of the blue venues to attract agents leads to significantly different levels of density around the venues of each group.

**Figs 13 and 14F–14H intolerant agents associated with non-exclusive venues can be repulsed from an area, leading them to "evacuate" the neighborhood and opening it to "co-optation" by the other group.**   In cases of high intolerance or extremely low exclusivity it is possible that some adventurous agents from another group can "co-opt" an "evacuated" venue (see Table 1). "Evacuation" refers to the abrupt departure of individuals from an area surrounding a venue of the same group. This results from a combination of relatively high intolerance amongst the individuals and the presence of at least a small number of individuals from the other group who are sufficiently adventurous to attend the venue. Their attendance at the venue makes the area intolerable for the other individuals at the venue. "Evacuation" leads to "co-optation" when the resulting vacancy encourages more individuals of the new group to arrive, and these individuals become the majority within the venue, even though it is associated to the other group that has departed.

Figs 13, 14G and 14H show how the interplay of venue "evacuation" and "cooptation" can change the composition of a neighborhood. In these scenarios where the blue agents are highly intolerant, the presence of some red agents at a blue venue can lead them to depart the area. Since in this scenario all of the blue agents within travel distance will visit a venue, their high intolerance, along with the presence of a few red agents at the venue, can cause the abrupt "evacuation" of the whole area by blue agents. In this situation, the venue acts as a kind of repulsive force keeping the area empty. When the local blue agents withdraw in this way, it is possible for the red agents to "co-opt" the venue and even attract more red agents from

elsewhere, but the number of vacant cells in the area generally remains quite high. In cases such as Figs 13 and 14H and S8 Video, especially when exclusivity is near 0, the area around a venue can flip entirely due to a slightly greater number of red agents in the initial distribution. These runs show additional ways in which opening up a venue to other groups can create surprising results that give insight into processes through which neighborhoods can turn-over from one group to another or cause an area to empty out entirely.

## Discussion

In the foregoing simulation experiments, we examined precisely defined parameters describing how venues interact with individual decisions to produce urban residential segregation and integration, and systematically varied their strength and interactions. Across our studies, it is clear that the outcomes of the Schelling model are largely dependent on the non-urban, venue-less environment in which they were run.

When venues are introduced, however, mixed outcomes become more likely in areas where agents can trade proximity to some neighbors for proximity to the group venue. Essentially, the parishioner is happier to live in a mixed neighborhood when they are close to a church, and the young single doesn't mind having older neighbors as long as there are bars full of other young people close by. However, as the venue catchment area grows and agents are no longer required to give up venue access to be near their neighbors, segregated outcomes become relatively more common.

We also showed that mixed outcomes among relatively intolerant agents are somewhat unstable. When venues are added to these simulations and wider and wider areas become viable for agents, they will tend to consolidate more into homogenous neighborhoods. In these cases, venues can act at a distance to generate segregation among those who do not visit them. Similarly, we show that similar venues can combine to have super-additive effects beyond their boundaries. In our radial and core/periphery models we demonstrate the importance of history: specifically venue location and the starting conditions of residents in determining the terminal level of segregation.

In Study 3, we show how a change in venue format can alter neighborhood segregation. We can think of this as representative of situations like school bussing, or the decision to allow women into men's clubs- situations where there is deliberate planning at the venue level. Study 3 also revealed an irony of urban life akin to the irony of social life revealed in Schelling's original model. Just as Schelling demonstrated that one cannot simply read individual intentions from collective patterns of behavior, our analysis shows that one cannot simply read organizational values from urban patterns of segregation. Relatively exclusive venues can generate diverse neighborhoods, while relatively inclusive venues can in some circumstances produce highly segregated areas.

Across these simulations, it is clear that the urban environment, and not just urban residents, plays a role in generating segregation. The urban order does still emerge partially with the sovereign decisions of homeowners. It is also, crucially, structured by venue interactions, where history has put venues, and what actions are undertaken at the venue level.

Whether it is the result of central planning, urban design, the decision making of individuals, or some combination of these factors, our research suggests that the nature and distribution of venues in cities could have a role to play in the development and stability of neighborhoods. Certain kinds and distributions of venues will encourage diverse urban communities while others could protect the comforts of shared values and collective identity. Evidently, the question of which is preferable exceeds the capacities of simulation and must lead instead to the normative considerations of an urban politics.

## Limitations and extensions

While this research has demonstrated that and how venues can cause spatially articulated patterns of segregation and integration to emerge, it is constrained by some limitations. These limitations are also opportunities for new research directions. For example, by treating venues as akin to what Epstein and Axtell call "the medium over which agents interact" [70:5], our simulation is limited to exploring a one-way relationship between venues and individual agents. Improving upon this and taking seriously the importance of circular causality in the simulation would not only lead to a better understanding of venues themselves; it would also allow us to build a richer notion of "neighborhood," which in the present model only denotes the set of nearby cells that an individual considers in evaluating the diversity of their surroundings. "Neighborhood" as presently defined is insufficient to support a discussion of the various and multi-scalar networks of participation and association that contribute to an individual's group identity. Communications technologies, of which online social networks are only the most recent iteration, imply that these networks need not be straightforwardly spatial at all. Moreover, our simulation model, which is configured in order to study segregation as a binary (two group) condition, has limited capabilities for considering the kinds of multi-group clustering and segregation that would emerge from a more robust conception of participation in both local neighborhoods and other, broader networks [71–73]. Further research that overcomes these limitations would contribute not only to the verisimilitude of the simulation, but also move toward a more robust heuristic model of urban segregation. Finally, while most of our examples and literature have been from the North American context, we believe that this phenomenon is more general, such as in the activity spaces in Belfast [74], Korean churches in Beijing, [75] or the "restaurants, cafes, markets, museums, [and] festivals where Germans and Turks exchange and meld their cultural values and practices" in Kreuzberg [76]. An important future direction is to more fully engage with a more international set of literatures and examples.

Because the research presented here is part of an ongoing project, we have sought to build a simulation model that can be developed and extended. In this sense, the means of overcoming many limitations are already embedded in the tool. Some extensions for further research have already been implemented in the model but for the sake of brevity and clarity are not explored in this paper. For example, the "manditoriness" relation between venue "obligatoriness" and individual "commitment" promises to complicate the simulation's dynamics in ways that are suggestive of actual venue attendance.

Other extensions can build upon the framework developed here to pursue more complex and dynamic processes. For example, although the work does not yet explore the possibility of more dynamic venues, at an implementation level both individuals and venues are represented as kinds of agent. In other words, in the underlying code of our simulation, both of these classes of object inherit from the same superclass, which could perhaps be considered as quasi-Latourian "actants" [77]. Because of this approach, future research can explore circular causalities by implementing venue processes such as re-locating, franchising, and closing, as well as succession and evolution based on their level of success. The interaction between individuals and venues could also be enriched by implementing a simple learning procedure, whereby the recurring act of visiting venues can feedback into each individual's sense of adventurousness and commitment. These extensions will also enrich the meaning of "neighborhoods," redefining them as outcomes of reciprocal relationships within the simulation space that can emerge, stabilize, and even collapse over time based on the kinds of generative mechanisms that are cataloged in Table 1. Additionally, future research can account for the complexity of individuals' participation in group identities and multiple simultaneous networks through venues by 1)

limiting individuals to a finite available time for visiting venues within each 'time-step', enabling a temporal segregation of venue-use for different subsets of visitors (i.e. "morning shift" and "evening shift") that may not interact, and 2) by changing the length of individuals' "Group" variable from a bit string of length one (enabling two groups) to a greater value that will enable multiple and potentially overlapping group identities [78]. An obvious but nontrivial development would be a model where venue location is endogenously determined.

As the number of parameters increases, we anticipate that our two-dimensional parameter-space representations will quickly prove insufficient as a means to explore and chart the interactions of different input values. Moving forwards, we will consider techniques for searching through high-dimensioned parameter spaces such as genetic algorithms as a replacement. We hope that this will enable us to continue identifying significant tipping point thresholds and domains of consistent results even without a brute-force exploration of every possible parameter combination. Additionally, we are interested to test the capacities of such computational techniques for model discovery driven by empirical data from real world cities [79].

The present study has made considerable progress toward understanding the role of venues in segregation dynamics. Many questions remain, however, especially about the role of various kinds of built form in neighborhood formation. Not least is the role of the built form of the venues themselves, including their architectural ornamentation and signage that can act as markers of collective visibility. For example, as Fong and Harold note (2017), the presence of a synagogue can provide a sense of community and cultural familiarity even among those who do not attend regularly–or become an object of scorn and resentment. Looking to Maurice Halbwachs [80], Aldo Rossi [81], and others could open up exciting research questions about how such legibility develops over time and constitutes a kind of collective memory through place-making activities, modifications to built form, and the introduction of new venues. Beyond physical venues, any number of other aspects of cities, including roads, walls, landscapes, but also zoning restrictions, housing policies, and variable land values would offer interesting subjects for future study [13, 14]. In each of these cases, the precise influence on segregation dynamics, and also the dynamic between the 'top-down' design of the built form and its 'bottom-up' occupation and use by groups of individuals are fruitful avenues to explore.

## Supporting information

**S1 Video.**
(TXT)

**S2 Video.**
(TXT)

**S3 Video.**
(TXT)

**S4 Video.**
(TXT)

**S5 Video.**
(TXT)

**S6 Video.**
(TXT)

**S7 Video.**
(TXT)

**S8 Video.**
(TXT)

**S1 Appendix. Complete settings for each study.**
(DOCX)

## Acknowledgments

This research was undertaken in the context of the "Urban Genome Project" at the University of Toronto. The authors wish to thank their collaborators in this project, especially Mark Fox, Rob Wright, and Fabio Dias. They are also grateful for feedback received at International Network of Analytical Sociologists conference in 2018 and from Yongren Shi and Angelina Grigoryeva.

## Author Contributions

**Conceptualization:** Daniel Silver, Ultan Byrne, Patrick Adler.

**Formal analysis:** Daniel Silver, Ultan Byrne.

**Investigation:** Daniel Silver, Ultan Byrne.

**Methodology:** Daniel Silver.

**Project administration:** Daniel Silver.

**Software:** Ultan Byrne.

**Validation:** Daniel Silver.

**Visualization:** Ultan Byrne.

**Writing – original draft:** Daniel Silver, Ultan Byrne, Patrick Adler.

**Writing – review & editing:** Daniel Silver, Ultan Byrne, Patrick Adler.

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
