## [Decision Letter · Decision Letter 0]

30 Jun 2020

PONE-D-20-13827

Venues and Segregation: A Revised Schelling Model

PLOS ONE

Dear Dr. Adler,

Thank you for submitting your manuscript to PLOS ONE. After careful consideration, we feel that it has merit but does not fully meet PLOS ONE’s publication criteria as it currently stands. Therefore, we invite you to submit a revised version of the manuscript that addresses the points raised during the review process.

We look forward to receiving your revised manuscript.

Kind regards,

Giorgio Fagiolo

Academic Editor

PLOS ONE

Additional Editor Comments:

Dear Authors, two reports are now in. I suggest to perform a major revision taking fully into account all the comments made by the reviewers, especially remarks and suggestions raised by Referee #1. Furthermore, it is necessary the code employed for simulation is made available in an online repository so as to allow for reproduction (see on this point a comment by Referee #2).

2. Thank you for stating the following in the Acknowledgments on the title page of your manuscript:

'Funding Information:

This project is funded under a Connaught Grant from the University of Toronto.'

'The author(s) received no specific funding for this work.'

3. Please include a caption for figure 13.

4. Please ensure that you refer to Figures 8 and 13 in your text as, if accepted, production will need this reference to link the reader to each figure.

Reviewers' comments:

Reviewer's Responses to Questions

**Comments to the Author**

1. Is the manuscript technically sound, and do the data support the conclusions?

Reviewer #1: Partly

Reviewer #2: Yes

2. Has the statistical analysis been performed appropriately and rigorously? 

Reviewer #1: N/A

Reviewer #2: Yes

3. Have the authors made all data underlying the findings in their manuscript fully available?

Reviewer #1: No

Reviewer #2: Yes

4. Is the manuscript presented in an intelligible fashion and written in standard English?

Reviewer #1: Yes

Reviewer #2: Yes

5. Review Comments to the Author

Reviewer #1: Manuscript “Venues and Segregation: A Revised Schelling Model” explores the role of urban venues in segregation. The authors attempt to expand the Schelling model by incorporating group-based venues for out-of-home social interactions. In doing so, the study joins effort to better understand urban segregation beyond residential neighborhoods, which is an important, if nascent, development in the segregation literature. The introduction of venues into the Schelling model is interesting and can potentially enhance the theoretical model of social sorting and segregation. I believe the manuscript can be substantially improved by addressing several issues.

First and foremost, the paper needs to make a better case for its methodology and explicitly state the basic assumptions early on. Only until further into the manuscript did I realize the authors treat venue-based exposures the same way as neighborhood-based exposures – individuals are satisfied as long as the percentage of “like” individuals in BOTH their neighborhoods and visited venues are above their tolerance level. This is a key assumption and needs to be clearly stated as soon as possible, and ideally justified by citations or theoretical considerations. What about the effect of a venue on adjacent residents of the other group? Even if they don’t visit the venue, wouldn’t they be somewhat exposed to the other group members that continuously enter their neighborhoods to visit the venue, which potentially make their neighborhoods more “mixed” and less satisfactory?

The literature review can be strengthened by recent studies examining segregation and social interactions in workplace, activity space, etc. If venues are used to incorporate out-of-home exposure, would it make sense to include job locations too, as these are both obligatory and independent of where one lives?

Why are the parameters (commitment, adventurousness, exclusivity, and obligatoriness) important? If I understand it correctly, an adventurous individual would be more likely exposed to individuals of the other group by visiting their venues. Does that mean they are less likely to be satisfied with their neighborhoods, wherever they live, unless they are out of the catchment areas of any venues of the other group? Or are adventurous individuals naturally more tolerant? I find this parameter adding little value and more unnecessary complexity to the model. Likewise the “obligatoriness” of the venues – in most of their models the authors test only a handful of venues. Obviously these are not your neighborhood grocery stores, theatres, etc., which likely have weaker association with group identities or sorting. Why not simply assume all venues tested are obligatory and exclusive?

The simulation models need to be better explained. How did the authors choose the patterns of venues they test in the three studies, and why? What are the main variations they are exploring and what are the key implications? The way the three studies are currently set up seems a bit arbitrary, with ad-hoc interpretations and “mechanisms.” What are “maximum travel distance” based on and do they vary by individual or venue? Similarly, how are commitment, adventurousness, exclusivity, and obligatoriness values set? What are the distributions? While a more adventurous individual is more likely to visit a venue of the other group how adventurous do they have to be (and how exclusive the venue has to be) for the visit to actually happen? Or is it based on a probability function?

Does a “locked in” state of highly intolerant agents actually happen in real life, or is it more of a limitation of the Schelling model? In reality intolerant agents may choose to move to more segregated locations even if those are still not “satisfactory,” or they can influence local policy to facilitate segregation. If the “locked in” state doesn’t necessarily happen in reality, what are the practical implications of venues “breaking” the “locked in” condition?

Table 2 is confusing. Are all four types of venues of the same group as the individual, or not? Maybe the information can be better illustrated on a two or three dimensional graph? Or just explain it clear enough in the main text so the table/figure won’t be needed?

The paragraph before “Findings” reads very confusing without a concrete example. What visualization method? What are the three studies about? What is “concordance” – is this a widely used term in the segregation literature? The authors need to either better define or justify the use of the term or to find a more accepted/comprehensible term. If it is an important measure it should be defined in the main text instead of an appendix.

The figures are very blurry. I couldn’t read A, B, and C in the first few figures.

The authors refer to the [1][2][3][4] in figures sometimes as a cell in the parameter space, sometimes as a full illustration of simulation patterns. There should be a way of distinguishing the two so readers don’t get lost. Label the figures with “locked in,” “bootstrap,” etc. if they are classified as such. The four patterns or mechanisms also need better definition and explanation.

Overall, I recommend a thorough editing of the manuscript and tightening up the language. Many sentences are confusing, repetitive, or too loose for a research article.

Reviewer #2: This manuscript extends the Schelling segregation model to account for the concept of “venues”. Venues are defined as the “areas where urbanities interact” and they are implemented mechanistically. The hypothesis is that venues play an important role in the decision-making processes leading to urban segregation. The authors conduct a series of simulations and report that venues make segregation less likely among relatively tolerant agents and more likely among the intolerant.

This manuscript explores an interesting idea on a really well-designed experimental study. The authors cite relevant literature, including the modern versions of the original segregation model produced by Benensen and Hatna. The authors methodology is correct and adequate for the exploration and the manuscript is well written.

Comment 1:

Causal inference of the mechanistic simulations: The authors could reinforce their methodology by using recent developments on the generative social science and agent-based modeling areas that could strengthen their claims for causality (eg., the explored mechanism “in simulation” have something meaningful to say about the real-world behavior). For instance, see recent workshop at:

https://www.igss-workshop.org

and a recent publication describing one such methodology at:

https://arxiv.org/abs/1802.00435

Comment 2:

The idea of Physical venues is very powerful, and the results are interesting and meaningful. In the COVID-19 era, it may be also interesting to extend the concept to “logical venues” like social media activity, virtual meetings and virtual groups. Would logical venues have any impact on physical segregation?

Recommendation:

Make all code and data used available for replication purposes along with all parameter value used on the experiments. One way is to publish your model at:

https://www.comses.net

6. PLOS authors have the option to publish the peer review history of their article (what does this mean?). If published, this will include your full peer review and any attached files.

Reviewer #1: No

Reviewer #2: No

---

## [Author Response · Author response to Decision Letter 0]

4 Sep 2020

To the Editor and Reviewers:

Thank you for the incisive review of our manuscript, “Venues and Segregation: a revised Schelling model.” We have revised the paper thoroughly in the light of the comments and believe it is now stronger. Below we describe how we responded to each point made by the academic editor and reviewers.

Responses to Reviewer 1:

Reviewer 1 raised a number of important points, primarily around clarifying some of our methodological assumptions. We have taken these to heart and completely revised the Methodology section of the manuscript. That section now has four sub-sections that describe 1) model parameters 2) model rules 3) analytics and 4) the experiments. Overall, this new Methodology section addresses most of the reviewer’s main concerns, while changes to other parts of the manuscript detailed below address the reviewer’s other comments. More specific responses are as follows:

Comment: “First and foremost, the paper needs to make a better case for its methodology and explicitly state the basic assumptions early on. Only until further into the manuscript did I realize the authors treat venue-based exposures the same way as neighborhood-based exposures – individuals are satisfied as long as the percentage of “like” individuals in BOTH their neighborhoods and visited venues are above their tolerance level. This is a key assumption and needs to be clearly stated as soon as possible, and ideally justified by citations or theoretical considerations.”

Response: Thank you for this comment. We now state this assumption in the Introduction, on p. 4 line 15, and in the section on “the value of incorporating venues,” p. 8 line 15 The remainder of that section offers theoretical justification and citations for including those encountered in venues and as neighbors in agents’ evaluation of their satisfaction with their locations. Finally, the simulation rules described on p. 16 line 27 describe the satisfaction calculation in precise terms. 

Comment: “What about the effect of a venue on adjacent residents of the other group? Even if they don’t visit the venue, wouldn’t they be somewhat exposed to the other group members that continuously enter their neighborhoods to visit the venue, which potentially make their neighborhoods more “mixed” and less satisfactory?”

Response: This is an interesting suggestion, and would involve incorporating into the model travel into/out of the neighbourhood to reach a venue. While intriguing, we believe that pursuing it here would add too much complexity to the model, though it could be worth including in future work. We have noted explicitly the simplifying assumption we make here, on p. 17 line 23: “ Since VSame and VTotal are only changed when A visits a given venue, the presence of the venue itself in a neighborhood does not influence agents’ contentment, nor are they affected by the process of others traveling into or out of the venue. This is a simplifying assumption in the present study that could be worth opening up in future work.”

Comment: The literature review can be strengthened by recent studies examining segregation and social interactions in workplace, activity space, etc. If venues are used to incorporate out-of-home exposure, would it make sense to include job locations too, as these are both obligatory and independent of where one lives?

Response: This is a good point. We have added citations and references to this issue on p. 3 line 12, and p 8, line 23. 

Comment: Why are the parameters (commitment, adventurousness, exclusivity, and obligatoriness) important? If I understand it correctly, an adventurous individual would be more likely exposed to individuals of the other group by visiting their venues. Does that mean they are less likely to be satisfied with their neighborhoods, wherever they live, unless they are out of the catchment areas of any venues of the other group? Or are adventurous individuals naturally more tolerant? I find this parameter adding little value and more unnecessary complexity to the model. Likewise the “obligatoriness” of the venues – in most of their models the authors test only a handful of venues. Obviously these are not your neighborhood grocery stores, theatres, etc., which likely have weaker association with group identities or sorting. Why not simply assume all venues tested are obligatory and exclusive?

Response. Thank you for these questions. We clarified our theoretical reasoning for including these parameters in the section “What makes a venue a venue?”. In doing so, we revised the parameter names to better indicate the theoretical logic. Specifically, we refer to “openness” and “mandatoriness” (pp 10-11) as more general categories, explain why they are important along with relevant citations, and then discuss how they each involve two aspects, one from the side of the venue (“exclusivity” and “obligatoriness”), the other from the side of the agent (“adventurousness” and “commitment”). We also clarify the theoretical distinction between “adventurousness” and tolerance (p. 12 line 26). In addition, the Methodology section defines the parameters in precise terms (pp 14 line 13 and pp 15 lines 30) and how they inform the agents’ decision rules (p 17 line 5). Methodologically, adventurousness and commitment are necessary to determine what agents visit venues that are not completely exclusive or obligatory, which in our view is a theoretically important situation to be able to model, since venues do range from cafes and grocery stores to orthodox churches, as discussed in the literature review. In practice, as we note on p. 16 line 4, adventurousness and commitment only matter in situations when exclusivity or obligatoriness are less than 1 or greater than 0, meaning that they do not come into play for Studies 1 or 2. Study 3 however is meant in part to illustrate how when exclusivity varies, adventurousness can sometimes create interesting and illuminating patterns of segregation/integration (p. 22 line 11, p. 31 line 22). Finally, we do include some parameters in the model that we do not explicitly explore in this paper, specifically obligatoriness and commitment. We do this for the sake of simplicity and brevity, but believe they should still be included in the framework for the theoretical reasons noted in the literature review (p 12 line 9). We explicitly note this now on p. 16, line 8, and on p. 38 line 14 discuss further exploration of obligatoriness/commitment as an important extension of the research agenda laid out in the present paper. 

Comment: “The simulation models need to be better explained. How did the authors choose the patterns of venues they test in the three studies, and why? What are the main variations they are exploring and what are the key implications? The way the three studies are currently set up seems a bit arbitrary, with ad-hoc interpretations and “mechanisms.” What are “maximum travel distance” based on and do they vary by individual or venue? Similarly, how are commitment, adventurousness, exclusivity, and obligatoriness values set? What are the distributions? While a more adventurous individual is more likely to visit a venue of the other group how adventurous do they have to be (and how exclusive the venue has to be) for the visit to actually happen? Or is it based on a probability function?”

Response: The Methodology section now defines the parameters and rules in more precise terms, including travel distance and the distributions of values for commitment, adventurousness, exclusivity, and obligatoriness. It also describes (p. 17 line 7) how exclusivity and adventurousness interact to determine if a given agent visits a given venue of the other group. The section “Simulation Experiments” (p. 19) describes the rationale for the patterns of venues we test with relevant citations as well as the main variations we explore. As we note on p. 23 line 9, we discovered various generative mechanisms in the course of conducting the experiments and so we note them in the course of the analysis. Nevertheless, in this revision we have attempted to more explicitly organize the discussion of findings in terms of a) patterns of segregation/integration and b) the mechanisms we uncovered that produce them. 

Comment: “Does a “locked in” state of highly intolerant agents actually happen in real life, or is it more of a limitation of the Schelling model? In reality intolerant agents may choose to move to more segregated locations even if those are still not “satisfactory,” or they can influence local policy to facilitate segregation. If the “locked in” state doesn’t necessarily happen in reality, what are the practical implications of venues “breaking” the “locked in” condition?”

Response: These are interesting questions. As we note on p. 25 lines 11, we believe that this sort of locking in has real-world analogs, for instance in cases of expensive urban land markets, and we cite relevant literature to this effect.

Comment: “Table 2 is confusing. Are all four types of venues of the same group as the individual, or not? Maybe the information can be better illustrated on a two or three dimensional graph? Or just explain it clear enough in the main text so the table/figure won’t be needed?”

Response: Thank you for this comment. We have indeed dropped Table 2 (as well as Table 1), as we feel the revised Methodology section and Literature Review explain them clearly enough. 

Comment: “The paragraph before “Findings” reads very confusing without a concrete example. What visualization method? What are the three studies about? What is “concordance” – is this a widely used term in the segregation literature? The authors need to either better define or justify the use of the term or to find a more accepted/comprehensible term. If it is an important measure it should be defined in the main text instead of an appendix.”

Response: The revised Methodology now describes the analytics we use, including the visualization techniques and the Concordance measure, in more detail, in the section “Simulation Analytics.” The section also discusses precedents for the Concordance measure in Schelling’s original article and why we choose to use this term.

Comment: “The figures are very blurry. I couldn’t read A, B, and C in the first few figures.”

Response: We have attempted to improve the legibility of the figures.

Comment: “The authors refer to the [1][2][3][4] in figures sometimes as a cell in the parameter space, sometimes as a full illustration of simulation patterns. There should be a way of distinguishing the two so readers don’t get lost. Label the figures with “locked in,” “bootstrap,” etc. if they are classified as such. The four patterns or mechanisms also need better definition and explanation.”

Response: The section in the Methodology on “Simulation Analytics” includes a new figure describing how to read the visualizations, including the labeling. There we note that the numeric labels in the parameter space visualizations (e.g. [1], [2], [3]) are in fact tied to the full illustrations of the specific simulation outcomes referred to with the same labels. This technique is important, as it enables us to refer back and forth between the overall space and then specific results within it. They are in fact referring to one and the same thing, so we do not distinguish between them. We have also labeled figures with relevant mechanisms where appropriate and defined each mechanism the first time it is mentioned, referring the reader also to Table 1, where all the mechanisms are compiled and defined. 

Comment: Overall, I recommend a thorough editing of the manuscript and tightening up the language. Many sentences are confusing, repetitive, or too loose for a research article.

Response: We have done our best to tighten and streamline the article. This led us to drop what was Study 1.2 from the original submission, which we believe has led to a more focused contribution. In addition, we have reorganized the results section so that it speaks directly to study outcomes and moved all methodological material (previously under results) to the pre-results section.

Responses to Reviewer 2:

Overall, Reviewer 2 was positive about the manuscript, noting it pursues an “interesting idea on a really well-designed experimental study. The authors cite relevant literature, including the modern versions of the original segregation model produced by Benensen and Hatna. The authors methodology is correct and adequate for the exploration and the manuscript is well written.”

The reviewer did make two important comments, to which we respond below:

Comment: “Causal inference of the mechanistic simulations: The authors could reinforce their methodology by using recent developments on the generative social science and agent-based modeling areas that could strengthen their claims for causality (eg., the explored mechanism “in simulation” have something meaningful to say about the real-world behavior). For instance, see recent workshop at:

https://www.igss-workshop.org

and a recent publication describing one such methodology at:

https://arxiv.org/abs/1802.00435”

Response: Thank you for this comment. This is an important direction to pursue, and we are doing so in our ongoing research. We note on p x, line y this extension and cite the relevant literature noted by the reviewer.

Comment: “The idea of Physical venues is very powerful, and the results are interesting and meaningful. In the COVID-19 era, it may be also interesting to extend the concept to “logical venues” like social media activity, virtual meetings and virtual groups. Would logical venues have any impact on physical segregation?”

Response: This is a very intriguing suggestion. We note the potential for the concept of venue to be extended to such settings on p. x, line y. 

Comment: “Make all code and data used available for replication purposes along with all parameter value used on the experiments. One way is to publish your model at:

https://www.comses.net”

Response: Thank you for your suggestion. Following it, and the editor's comment, we have made the Java code used for our simulation model available at comses.net. The code can be accessed at: https://www.comses.net/codebases/122f9b9f-4aeb-419e-afdd-a7e211f62dfb/releases/1.0.0/ under the title "Venues and Segregation, A Revised Schelling Model"

---

## [Decision Letter · Decision Letter 1]

22 Sep 2020

PONE-D-20-13827R1

Venues and Segregation: A Revised Schelling Model

PLOS ONE

Dear Dr. Adler,

Thank you for submitting your manuscript to PLOS ONE. After careful consideration, we feel that it has merit but does not fully meet PLOS ONE’s publication criteria as it currently stands. Therefore, we invite you to submit a revised version of the manuscript that addresses the points raised during the review process.

We look forward to receiving your revised manuscript.

Kind regards,

Giorgio Fagiolo

Academic Editor

PLOS ONE

Reviewers' comments:

Reviewer's Responses to Questions

**Comments to the Author**

1. If the authors have adequately addressed your comments raised in a previous round of review and you feel that this manuscript is now acceptable for publication, you may indicate that here to bypass the “Comments to the Author” section, enter your conflict of interest statement in the “Confidential to Editor” section, and submit your "Accept" recommendation.

Reviewer #2: All comments have been addressed

Reviewer #3: All comments have been addressed

2. Is the manuscript technically sound, and do the data support the conclusions?

Reviewer #2: Yes

Reviewer #3: Yes

3. Has the statistical analysis been performed appropriately and rigorously? 

Reviewer #2: Yes

Reviewer #3: Yes

4. Have the authors made all data underlying the findings in their manuscript fully available?

Reviewer #2: Yes

Reviewer #3: Yes

5. Is the manuscript presented in an intelligible fashion and written in standard English?

Reviewer #2: Yes

Reviewer #3: Yes

6. Review Comments to the Author

Reviewer #2: The authors have adequately addressed all my suggestions and modified the manuscript accordingly. They have also published their code as suggested.

Reviewer #3: I have not previously assessed this paper, presenting the integration of venues within Shelling's self-segregation model. The paper is well written. I do think that the authors have replied to previous reviewer comments and they have also made their code available (a MUST for any modeling paper). Theoretically, it would be great if the authors could upload an ODD protocol together with the model code. The Overview Design Details protocol is a fairly standard way to document agent based models. (ODD: Grimm, V., Berger, U., DeAngelis, D. L., Polhill, J. G., Giske, J., & Railsback, S. F. (2010). The ODD protocol: a review and first update. Ecological modelling, 221(23), 2760-2768.)

Having stated that, there are a few minor details that I think will need to be addressed before the paper can be published:

First and foremost, I would strongly suggest the authors to revise their "figure references" they are odd, very hard to follow and I think, in places, quite incorrect. For example Fig. 6 referred to in the text should be Fig. 3 (that is the figure showcasing two venues, one south and one north). Also, when you refer to the figure panels, you should always refer to the figure itself. For example [1].[2].[3] etc.. should actually be Fig X panels 1,2,3 through 6) or something similar. If not, a reader is left to wonder what those numbers are.

Second, but this I probably because I am not from the US, venues and segregation and the work you are doing is typical in the US but not necessarily elsewhere. I think this may be an important clarification. Unsure if European cities are similar or, for that matters, other places in the world.

Minor issues:

page 3 Line 24, I would just state either ABM or computational model instead of we developed a computer simulation, agent-based model.

page 4 line 11: not sure what you mean by deep irony.

page 5 line 8: not sure what higher levels refers to here.

Figures: Make sure you add labels that are legible. Increase font and in case, add them slightly outside the moore neighborhood you take into account. As of now, figures are not of publicatoin quality.

Page 9 line 5: This is typical for the US much less so for other countries. I would clarify that.

Page 15: when you refer to random distribution, do you actually refer to a random-uniform distribution? random-normal? random-Poisson? It is important to clarify the type of the distribution. I suspect it being random uniform, if so, please state so.

Page 16: Simulation runs: It would be really helpful to have a figure representing the scheduling of the a typical time-step in the simulation run (i.e. a flowchart of the tuypical simulatoin run, setup -> agent look around -=> agent decide to go to venue -> calculate propportoin -> agent satisfied? -> stopping conditions. See the ODD protocol in the reference above).

Page 17 one 8: The fact that agents can visit multiple venues at the time is an interesting assumption. That means that A can visit all venues in one time-step... is there any justification as to why this would be the case instead of being limited to visit 1 venue at each time-step?

Page 18 Line 15: I would eliminate the "rather than" running one simulation run per parameter combination with an ABM is flat out wrong, as one need to account for the probabilistic events (and overall model stochasticity) hence, a run is not representative enough of the model behavior under a specific parameter configuration.

Page 18 Line 19: 10 runs per parameter combination seems to be very a very low number... how much stochasticity is in the model? Normally i would expect this number to be at least 30 or, better yet, untill the standard error is neglgible. In past studies i have seen parameter runs in between 30 and 100 or 1000 depending on the variability and the amount of stochasticity inbuild in the model.

However, you also indicate a very low variability between runs, I would add that here as well to validate your choice of doing 10simulation run per parameter combination.

Also Page 18 Line 21: I wonder why the selection of 20 time-steps. Seems rather a short time, is this the only stopping condition? (often one of the stopping condition is until agent can not move).

Page 19 line 21: this looks very similar to the polarization metric used in Montalvo and Reynal-Quierol as well as in Baggio and Papyrakis. No need to cite, just for reference. ( Montalvo, J. G., & Reynal-Querol, M. (2005). Ethnic polarization, potential conflict, and civil wars. American economic review, 95(3), 796-816. and Baggio, J. A., & Papyrakis, E. (2010). Ethnic diversity, property rights, and natural resources. The Developing Economies, 48(4), 473-495.)

Page 23 Line 17: the figures seem to show 6 venues, and figures should be 5 and 6, not 6 and 7. The figure that shows only 2 venues is actually figure 3. The confusion on figures and labels makes the rest of the results quite hard to follow. Also in line 21: the venue distribution in figure 7 is more mixed... i think the author have the figures mixed up.

Page 25 Line 16: what do these numbers represent? ([2,3,4] I guess these are figure panels, but which figure? also may be best to label panels with letters.

Page 40, Line 24: If this is work, you should cite it appropriately.

7. PLOS authors have the option to publish the peer review history of their article (what does this mean?). If published, this will include your full peer review and any attached files.

Reviewer #2: No

Reviewer #3: No

---

## [Author Response · Author response to Decision Letter 1]

27 Oct 2020

To the Editor and Reviewers:

Thank you for the incisive review of our manuscript, “Venues and Segregation: a revised Schelling model.” We are gratified that both reviewers judged the paper to be technically sound, that all analyses had been performed appropriately and rigorously, that all underlying data are publicly available, that the writing is clear, and that all comments from previous revisions were addressed. Reviewer #3 raised a few additional “minor details.” We describe how we addressed these below.

Comment: “I would strongly suggest the authors to revise their "figure references" they are odd, very hard to follow and I think, in places, quite incorrect.” 

Response: thank you for this comment. We have clarified our figure references in a number of ways. First, the captions to Figures 5 and 6 provide detailed explanations of our labeling convention. To make the convention more perspicuous, we use bracketed letters (e.g. [A]) rather than numbers now. Second, we also remind the reader of this convention in all relevant figure captions. For example, the caption to Figures 7 and 8 explain how the letters in Figure 7 refer to specific simulation runs illustrated in Figure 8, and the caption in Figure 8 explains how the letters in Figure 8 can be used to locate those specific results in the larger parameter space shown in in Figure 7. Third, when referring in the text to the specific runs indicated by the lettering system described in the captions to Figures 5 and 6, we specify both the figure number and appropriate letter. For example, on p. 24 line 25 we refer to “Figs 7-8 [A].” This indicates that we are discussing the run labelled with the red “A” in the parameter space shown in Figure 7, which corresponds to the result shown in the panel labeled [A] in Figure 8. 

Comment: “this I probably because I am not from the US, venues and segregation and the work you are doing is typical in the US but not necessarily elsewhere. I think this may be an important clarification. Unsure if European cities are similar or, for that matters, other places in the world.”

Response: This is an interesting point. We take the point that most of our examples and the literature we cite mostly speaks to the North American context. That said, we believe that venues play a role in segregation and integration elsewhere. We provide three quick examples on p. 39, lines 10-15, though we note that fully engaging with more international examples and literatures is an important direction for future research. 

Other minor points:

“page 3 Line 24, I would just state either ABM or computational model instead of we developed a computer simulation, agent-based model.”

Response: done

page 4 line 11: not sure what you mean by deep irony.

Response: added a definition of the notion of irony as used in social theory, based on Brown 1978.

page 5 line 8: not sure what higher levels refers to here.

Response: added “of tolerance.”

Figures: As of now, figures are not of publicatoin quality.

Response: we increased the font size and resolution of the figures, which we hope are now legible.

Page 9 line 5: This is typical for the US much less so for other countries. I would clarify that.

Response: added a clarification

Page 15: when you refer to random distribution, do you actually refer to a random-uniform distribution? random-normal? random-Poisson? It is important to clarify the type of the distribution. I suspect it being random uniform, if so, please state so.

Response: changed to “random uniform”

Page 16: Simulation runs: It would be really helpful to have a figure representing the scheduling of the a typical time-step in the simulation run (i.e. a flowchart of the tuypical simulatoin run, setup -> agent look around -=> agent decide to go to venue -> calculate propportoin -> agent satisfied? -> stopping conditions. See the ODD protocol in the reference above).

Response: added a flow chart as figure 3. 

Page 17 one 8: The fact that agents can visit multiple venues at the time is an interesting assumption. That means that A can visit all venues in one time-step... is there any justification as to why this would be the case instead of being limited to visit 1 venue at each time-step?

Response: Added a note on p. 17 lines 11-15 explaining this assumption. 

Page 18 Line 15: I would eliminate the "rather than" running one simulation run per parameter combination with an ABM is flat out wrong, as one need to account for the probabilistic events (and overall model stochasticity) hence, a run is not representative enough of the model behavior under a specific parameter configuration.

Response: done.

Page 18 Line 19: 10 runs per parameter combination seems to be very a very low number... how much stochasticity is in the model? Normally i would expect this number to be at least 30 or, better yet, untill the standard error is neglgible. In past studies i have seen parameter runs in between 30 and 100 or 1000 depending on the variability and the amount of stochasticity inbuild in the model.

However, you also indicate a very low variability between runs, I would add that here as well to validate your choice of doing 10simulation run per parameter combination.

Response: added a comment on this on p. 18 lines 28-32.

Also Page 18 Line 21: I wonder why the selection of 20 time-steps. Seems rather a short time, is this the only stopping condition? (often one of the stopping condition is until agent can not move).

Response: added a comment on this on p. 19, lines 2-4. As we note there, in none of the cases evaluated for this paper were there still agents willing and able to move by the twentieth time step. 

Page 19 line 21: this looks very similar to the polarization metric used in Montalvo and Reynal-Quierol as well as in Baggio and Papyrakis. No need to cite, just for reference. ( Montalvo, J. G., & Reynal-Querol, M. (2005). Ethnic polarization, potential conflict, and civil wars. American economic review, 95(3), 796-816. and Baggio, J. A., & Papyrakis, E. (2010). Ethnic diversity, property rights, and natural resources. The Developing Economies, 48(4), 473-495.)

Response: thank you for this information, which is very interesting. 

Page 23 Line 17: the figures seem to show 6 venues, and figures should be 5 and 6, not 6 and 7. The figure that shows only 2 venues is actually figure 3. The confusion on figures and labels makes the rest of the results quite hard to follow. Also in line 21: the venue distribution in figure 7 is more mixed... i think the author have the figures mixed up.

Page 25 Line 16: what do these numbers represent? ([2,3,4] I guess these are figure panels, but which figure? also may be best to label panels with letters.

Response: for this and the previous comment, please see our response regarding our labeling conventions, which should clarify this concern. 

Page 40, Line 24: If this is work, you should cite it appropriately.

Response: done. 

Thank you, once again, for your consideration of this piece.

Best

The Authors

---

## [Decision Letter · Decision Letter 2]

6 Nov 2020

Venues and Segregation: A Revised Schelling Model

PONE-D-20-13827R2

Dear Dr. Adler,

We’re pleased to inform you that your manuscript has been judged scientifically suitable for publication and will be formally accepted for publication once it meets all outstanding technical requirements.

Kind regards,

Giorgio Fagiolo

Academic Editor

PLOS ONE

Additional Editor Comments (optional):

Reviewers' comments:

Reviewer's Responses to Questions

**Comments to the Author**

1. If the authors have adequately addressed your comments raised in a previous round of review and you feel that this manuscript is now acceptable for publication, you may indicate that here to bypass the “Comments to the Author” section, enter your conflict of interest statement in the “Confidential to Editor” section, and submit your "Accept" recommendation.

Reviewer #3: All comments have been addressed

2. Is the manuscript technically sound, and do the data support the conclusions?

Reviewer #3: Yes

3. Has the statistical analysis been performed appropriately and rigorously? 

Reviewer #3: Yes

4. Have the authors made all data underlying the findings in their manuscript fully available?

Reviewer #3: Yes

5. Is the manuscript presented in an intelligible fashion and written in standard English?

Reviewer #3: Yes

6. Review Comments to the Author

Reviewer #3: The authors have replied to all my previous (minor) queries. The article is, in my opinion, ready for publication.

7. PLOS authors have the option to publish the peer review history of their article (what does this mean?). If published, this will include your full peer review and any attached files.

Reviewer #3: No

---

## [Editor Report · Acceptance letter]

10 Nov 2020

PONE-D-20-13827R2 

Venues and Segregation: A Revised Schelling Model 

Dear Dr. Adler:

I'm pleased to inform you that your manuscript has been deemed suitable for publication in PLOS ONE. Congratulations! Your manuscript is now with our production department. 

Kind regards, 

on behalf of

Dr. Giorgio Fagiolo 

Academic Editor

PLOS ONE